# Exploring accumulation-mode-$H_2SO_4$ versus $SO_2$ stratospheric sulfate geoengineering in a sectional aerosol-chemistry-climate model

Sandro Vattioni[1,2], Debra Weisenstein[2], David Keith[2], Aryeh Feinberg[1], Thomas Peter[1], Andrea Stenke[1]

[1]Institute of Atmospheric and Climate Science, ETH Zürich, Zurich, 8092, Switzerland
[2]Harvard John A. Paulson School of Engineering and Applied Sciences, Cambridge, MA-02138, USA

*Correspondence to:* Sandro Vattioni (sandro.vattioni@outlook.com)

**Abstract.** Stratospheric sulfate geoengineering (SSG) could contribute to avoiding some of the adverse impacts of climate change. We used the global aerosol-chemistry-climate model SOCOL-AER to investigate 21 different SSG scenarios, each with 1.83 Mt S $yr^{-1}$ injected either in form of accumulation-mode-$H_2SO_4$ droplets (AM-$H_2SO_4$), gas-phase $SO_2$, or as
combinations of both. For most scenarios, the sulfur was continuously emitted at 50 hPa ($\approx$ 20 km) altitude in the tropics and subtropics. We assumed emissions to be zonally and latitudinally symmetric about the equator. The spread of emissions ranges from 3.75° S-3.75° N to 30° S-30° N. In the $SO_2$ emission scenarios, continuous production of tiny nucleation mode particles results in increased coagulation, which together with gaseous $H_2SO_4$ condensation produces coarse mode particles. These large particles are less effective for backscattering solar radiation and have a shorter stratospheric residence time than
AM-$H_2SO_4$ particles. On average, the stratospheric aerosol burden and corresponding all sky short-wave radiative forcing for the AM-$H_2SO_4$ scenarios are about 37 % larger than for the $SO_2$ scenarios. Simulated stratospheric aerosol burdens show a weak dependence on the latitudinal spread of emissions. Emitting 30° N-30° S instead of 10°N-10°S decreases stratospheric burdens only by about 10 %. This is because a decrease in coagulation and the resulting smaller particle size is roughly balanced by faster removal through stratosphere-to-troposphere transport via tropopause folds. Increasing the injection
altitude is also ineffective, although it generates a larger stratospheric burden, because enhanced condensation and/or coagulation leads to larger particles, which are less effective scatterers. In the case of gaseous $SO_2$ emissions, limiting the sulfur injections spatially and temporally in the form of point and pulsed emissions reduces the total global annual nucleation, leading to less coagulation and thus smaller particles with increased stratospheric residence times. Pulse or point emissions of AM-$H_2SO_4$ have the opposite effect: they decrease stratospheric aerosol burden by increasing coagulation and
only slightly decreased clear sky radiative forcing. This study shows that direct emission of AM-$H_2SO_4$ results in higher radiative forcing for the same sulfur equivalent mass injection strength than $SO_2$ emissions, and that the sensitivity to different injection strategies vary for different forms of injected sulfur.

# 1 Introduction

Driven by human emissions, long-lived atmospheric greenhouse gas (GHG) concentrations now exceed levels ever experienced by *Homo sapiens*. The effects of these GHGs—as written by the Intergovernmental Panel on Climate Change in 2014—"have been detected throughout the climate system and are extremely likely to have been the dominant cause of the observed warming since the mid-20[th] century" (IPCC, 2014). Emissions of $CO_2$ and other GHGs must be curbed to reduce the impacts of climate change. Yet, the long lifetime of $CO_2$ and some other GHGs suggest that even if emissions were eliminated today, climate change and resulting human and environmental risks would persist for centuries.

We might bring global warming to a halt or reduce its rate of growth by combining emissions cuts with other interventions, such as a deliberate increase in the Earth's stratospheric aerosol burden, which would enhance the albedo of the stratospheric aerosol layer and reduces solar climate forcing. This idea, now often called "solar geoengineering", "solar climate engineering" or "solar radiation management", was first proposed by Budyko (1977) who suggested injecting sulfate aerosols into the stratosphere to increase Earth's albedo. Research on this topic became tabooed because of the entailed risks. However, efforts were renewed after Crutzen (2006) suggested that solar radiation management might be explored as a useful climate change mitigation tool, since adequate emission reductions were becoming increasingly unlikely.

Most research on solar geoengineering has focused on stratospheric sulfate geoengineering (SSG) through $SO_2$ injection, in part due to its volcanic analogues such as the 1991 eruption of Mt. Pinatubo. Studies of SSG, however, have found limitations with $SO_2$ injection as a method of producing a radiative forcing (RF) perturbation. Limitations include: (1) reduced efficacy at higher loading, limiting the achievable short-wave radiative forcing (Heckendorn et al., 2009; Niemeier et al., 2011; English et al., 2012; Niemeier and Timmreck, 2015; Kleinschmitt et al., 2018) (2) increased lifetimes of methane and other GHGs (Visioni et al., 2017; Tilmes et al., 2018), (3) impacts on upper tropospheric ice clouds (Kuebbeler et al., 2012; Visioni et al., 2018b) and (4) stratospheric heating (Heckendorn et al., 2009; Ferraro et al., 2011), especially in the tropical lower stratosphere, which would modify the Brewer-Dobson circulation (Brewer, 1949; Dobson, 1956) and increase stratospheric water vapor. Limitation (1) is primarily a function of the sulfate particle size distribution, determining their gravitational removal, whereas (2) to (4) are primarily dependent on chemical and radiative particle properties.

The size distribution problem with $SO_2$ injection arises after oxidation of $SO_2$ to $H_2SO_4$ when aerosol particles are formed through nucleation and condensation. Condensation onto existing particles increases their average size. In addition, the continuous flow of freshly nucleated particles leads to coagulation, both through self-coagulation of the many small new particles and—more importantly—coagulation with already existing bigger particles from the background aerosol layer. These particles then grow further through coagulation and condensation, which increases the average sedimentation velocity of the aerosol population (Heckendorn et al., 2009). Mean particle sizes tend to increase with the $SO_2$ injection rate, reducing the stratospheric aerosol residence time and, hence, their radiative forcing efficacy (e.g., W m$^{-2}$ (Mt S yr$^{-1}$)$^{-1}$). This problem could be reduced—and the radiative efficacy increased—if there was a way to produce additional accumulation mode (0.1-1.0 µm radius) sulfate particles (AM-$H_2SO_4$). Such particles are sufficiently large to decrease their mobility and hence their

coagulation. Furthermore, such particles are close to the radius of maximum mass specific up-scattering of solar radiation on sulfate particles, which is ~0.3 µm (Dykema et al., 2016). One proposed method of doing this is to directly inject $H_2SO_4$ vapor into a rapidly expanding aircraft plume during stratospheric flight, which would be expected to lead to the formation of accumulation mode particles with a size distribution that depends on the injection rate and the expansion characteristics of

the plume (Pierce et al., 2010). Two theoretical studies, Pierce et al. (2010) and Benduhn et al. (2016), suggest that appropriate size distributions could be produced in aircraft plumes using this method.

        To evaluate a geoengineering approach with AM-$H_2SO_4$ one needs to study the evolution of aerosol particles after injection of $H_2SO_4$ vapor into an aircraft wake and the subsequent transport and evolution of the aerosol plume around the globe. This is a problem with temporal scales ranging from milliseconds to years, and spatial scales from millimeters to

thousands of kilometers. At present there is no model that could seamlessly handle the entire range. However, the problem can be divided into two separate domains: (a) from injection to plume dispersal, and (b) from plume dispersal to global scale distribution. Each domain has associated uncertainties, but these can be studied separately with different modeling tools: plume dispersion models for (a), and general circulation models (GCMs) or chemistry-climate models (CCMs) for (b).

        *(a) Plume modeling.* The integration of the plume model starts with the production of small particles in a plume

from the exit point of the injection nozzle, and ends when the plume has expanded sufficiently so that the loss of particles by coagulation with ambient particles dominates the self-coagulation, whereupon the GCM or CCM becomes the appropriate tool (Pierce et al., 2010). The plume model needs to account for the initial formation of nucleation mode particles below 0.01 µm radius by homogeneous nucleation of $H_2SO_4$ and $H_2O$ vapor and the subsequent evolution of the particle size distribution by coagulation of the nucleation mode, as well as by condensation of $H_2SO_4$ vapor on already existing particles. In an

expanding aircraft plume, these processes occur on timescales from milliseconds to hours, and length scales from millimeters to kilometers. This was addressed by Pierce et al. (2010) and then by Benduhn et al. (2016). There is rough agreement that particles of 0.095-0.15 µm radius could be produced after the initial plume processing, but these results are subject to uncertainties and need further investigation.

        *(b) General circulation modeling.* The second part of the problem can be analysed by a GCM or a CCM, starting

from the release of sulfate particles of the size distribution calculated by the plume model into the grid of the GCM, all the way to implications on aerosol burden, radiative forcing, ozone, stratospheric temperature and circulation. To this end, the GCM must be coupled to chemistry and aerosol modules. The GCM then provides solutions on how the new accumulation mode particles change the large-scale size distribution, and thus the overall radiative and dynamical response to sulfate aerosol injection. Missing in this methodology are processes smaller than the grid size of the GCM, which may involve

filaments of injected material being transported in thin layers. Consideration of these sub-grid scale processes remains an uncertainty of our study, but might be handled by a Lagrangian transport model in a future study.

        A sectional or also called size-bin resolved aerosol module is important for a mechanistic understanding of the factors that determine the size distributions of the aerosols. Sectional aerosol models handle the aerosols in different size

bins (40 in SOCOL-AER) whereas modal models usually only apply 3 modes (e.g. Niemeier et al., 2011; Tilmes et al., 2017), each with different mode radius ($r_m$) and fixed distribution widths ($\sigma$), to describe the aerosol distribution. Therefore, the degrees of freedom among modal models usually is 3, whereas there are 40 for a sectional model like SOCOL-AER. Thus, sectional aerosol models represent aerosol distributions with better accuracy, though numerical diffusion does result

from the discretization in size space. Two earlier studies of SSG modelling, Heckendorn et al. (2009) and Pierce et al. (2010), used the AER-2D chemistry-transport-aerosol model with sectional microphysics (Weisenstein et al., 1997, 2007). Although the sectional aerosol module within has high size resolution, this 2D model only has a limited spatial resolution with simplified dynamical processes. So far, four different GCM models were used to study SSG with sectional aerosol modules, namely English et al. (2012), Laakso et al. (2016, 2017), Visioni et al. (2017, 2018a, 2018b) and Kleinschmitt et al.

(2018). English et al. (2012) used the GCM WACCM (Garcia et al., 2007) coupled to the sectional aerosol module CARMA (Toon et al., 1988) to simulate various SSG scenarios with sulfur emissions in the form of $SO_2$-gas, $H_2SO_4$-gas and AM-$H_2SO_4$, but without treatment of the quasi biennial oscillation (QBO) and without online interaction between aerosols, chemistry and radiation. The three other studies (Laakso et al., 2016, 2017; Visioni et al., 2017, 2018a, 2018b; Kleinschmitt et al., 2018) only performed $SO_2$ emission scenarios, but no AM-$H_2SO_4$ emission scenarios. Laakso et al. (2016, 2017) used

the GCM MA-ECHAM5 interactively coupled to the sectional aerosol module HAM-SALSA (Kokkola et al., 2008; Bergman et al., 2012). However, in both studies stratospheric chemistry was simplified using prescribed monthly mean OH and ozone concentrations. The ULAQ-CCM, which was used in Visioni et al. (2017, 2018a, 2018b) includes an interactive sectional aerosol module and additionally treats detailed stratospheric chemistry. Kleinschmitt et al. (2018) used the GCM LMDZ (Hourdin et al., 2006, 2013) which was coupled to the sectional aerosol module S3A (Kleinschmitt et al., 2017). In

their model setup, the aerosols were fully interactive with the radiative scheme, but the model included only simplified chemistry and a prescribed $SO_2$-to-$H_2SO_4$ conversion rate.

       There have been prior studies with other advanced interactive GCMs, but using modal aerosol schemes. Niemeier et al. (2011) looked at $SO_2$ and $H_2SO_4$ gas injection by using the GCM MA-ECHAM interactively coupled to the modal aerosol module HAM (Stier et al., 2005). Chemistry was simplified similar to Laakso et al. (2016, 2017) using prescribed OH and

ozone concentrations. Kravitz et al. (2017), MacMartin et al. (2017), Richter et al. (2017) and Tilmes et al. (2017) used the fully coupled global chemistry-climate model CESM1 (Hurrell et al., 2013; Mills et al., 2017) to simulate $SO_2$ emission scenarios. In their model setup, they applied higher horizontal and vertical resolutions compared to SOCOL-AER as well as a fully coupled ocean module and more complex chemistry. However, they also relied on a modal aerosol module, which in turn was coupled to cloud microphysics.

In this study we investigate different $SO_2$ and AM-$H_2SO_4$ emission scenarios by using the sectional global 3D aerosol-chemistry-climate model SOCOL-AER (Sheng et al., 2015), which treats prognostic transport as well as radiative and chemical feedbacks of the aerosols online in one model. As described above, GCMs are not yet able to interactively couple plume dispersion models. Hence, we follow Pierce et al., 2010 and use a log-normal distribution for the injected

aerosols in the AM-$H_2SO_4$ emission scenarios assuming that a certain size distribution can be created in an emission plume (Pierce et al., 2010; Benduhn et al., 2016). $SO_2$ emission scenarios are performed as a reference, and to gain insight into aerosol formation processes on a global scale. We perform a number of sensitivity studies with both $SO_2$ and AM-$H_2SO_4$ emissions that highlight differences between the two injection strategies and indicate future research needs.

## 2 Model Description

We use the global 3D aerosol-chemistry-climate model SOCOL-AER (Sheng et al., 2015) in this study. An earlier geoengineering study by Heckendorn et al. (2009) used SOCOLv2 (Egorova et al., 2005; Schraner et al., 2008) to study the stratospheric response to $SO_2$ injections. In that case, the AER–2D model (Weisenstein et al., 1997, 2007) was used to calculate global aerosol properties, which were prescribed in the 3D chemistry-climate model SOCOLv2. The SOCOL-AER (Sheng et al., 2015; Sukhodolov et al., 2018) model is based on SOCOLv3 (Stenke et al., 2013), and improves on the earlier versions of SOCOL by incorporating a sectional aerosol module based on the AER-2D model. In a recent study SOCOL-AER has been successfully applied to simulate the magnitude and the decline of the resulting aerosol plume after the 1991 Mt. Pinatubo eruption (Sukhodolov et al., 2018).

SOCOL-AER includes the chemistry of the sulfate precursors $H_2S$, $CS_2$, dimethyl sulfide ($C_2H_6S$, DMS), OCS, methanesulfonic acid ($CH_4O_3S$, MSA), $SO_2$, $SO_3$, and $H_2SO_4$, as well as the formation and evolution of particulates via particle size resolving microphysical processes such as homogeneous bimolecular nucleation of $H_2SO_4$ and $H_2O$, condensation and evaporation of $H_2SO_4$ and $H_2O$, coagulation and sedimentation. As opposed to an earlier version of SOCOL-AER that used a wet radius binning scheme (Sheng et al., 2015), this model version separates the aerosol according to $H_2SO_4$ mass in 40 bins, with $H_2SO_4$ mass doubling between neighboring bins. The new binning approach allows for more accurate consideration of size distribution changes caused by evaporation and condensation of $H_2O$ on the sulfate aerosols. Depending on the grid-box temperature and relative humidity, wet aerosol radii in the new scheme can range from 0.4 nm to 7 µm. Nevertheless, to simplify post-processing of results the sulfate aerosols are rebinned into the original wet size bins of Sheng et al. (2015). SOCOL-AER interactively couples the aerosol module AER with the chemistry module MEZON (Rozanov et al., 1999, 2001; Egorova et al., 2001, 2003) via photochemistry of the sulfate precursor gases as well as heterogeneous chemistry on the particle surfaces. In SOCOL-AER, MEZON treats 56 chemical species of oxygen, nitrogen, hydrogen, carbon, chlorine, bromine and sulfur families with 160 gas phase reactions, 58 photolysis reactions, and 16 heterogeneous reactions, representing the most relevant aspects of stratospheric chemistry. SOCOL-AER also treats tropospheric chemistry, though with a reduced set of organic chemistry (isoprene as most complex organic species), and prescribed aerosols (other than sulfate aerosols, which are fully coupled). SOCOL-AER also interactively couples AER with the general circulation model ECHAM5.4 (Manzini et al., 1997; Roeckner et al., 2003, 2006) of SOCOLv3 via the radiation scheme. SOCOL-AER treats 6 short-wave (SW) radiation bands between 185 nm and 4 µm as well as 16 long-wave radiation bands in the spectral range 10–3000 cm$^{-1}$. The extinction coefficients, which are required for each of the 22

wavelengths, as well as the single scattering albedo and the asymmetry factors, which are only taken into account for the 6 short-wave bands, are calculated from the particle size distribution of the 40 size bins according to Mie scattering theory (Biermann et al., 2000), with radiative indexes from Yue et al. (1994). The transport of the sulfur gas species and the aerosol bins is integrated into the advection scheme of ECHAM5 (Lin and Rood, 1996). MEZON is interactively coupled to
ECHAM5 using the three-dimensional fields of temperature, wind and radiative forcing of water vapor, methane, ozone, nitrous oxide and chlorofluorocarbons.

Operator splitting is used, whereupon transport is calculated every 15 minutes, and chemistry, microphysics and radiation are calculated every 2 hours, with 3-minute sub-timesteps for microphysical processes. We used T31 horizontal truncation (i.e. 3.75° resolution in longitude and latitude) with a vertical resolution of 39 hybrid sigma-p levels from the
surface up to 0.01 hPa (i.e. about 80 km altitude). This results in a vertical resolution of about 1.5 km in the lower stratosphere.

This study is among the first modelling studies on SSG which couple a size-resolved sectional aerosol module interactively to well-described stratospheric chemistry and radiation schemes in a global three-dimensional chemistry-climate model. Furthermore, this study explores the injection of AM-$H_2SO_4$ in detail and contrasts the resulting atmospheric
effects and sensitivities with those of gaseous $SO_2$ injections.

## 3. Experimental Setup

In this study, 21 different injection scenarios with annual emissions of 1.83 Mt of sulfur per year (Mt S yr$^{-1}$) in the form of AM-$H_2SO_4$ (sulfate aerosols) or gaseous $SO_2$ were performed, as well as runs with mixtures of both species. Per year, this corresponds to about 8-20 % of the sulfur emitted by the 1991 Mt. Pinatubo eruption, depending on the model under
consideration and the applied boundary conditions (Mills et al., 2016; Pitari et al., 2016; Sukhodolov et al., 2018; Timmreck et al., 2018). See Table 1 for a complete list of scenarios. Additionally, a reference run (termed "BACKGROUND") without artificial sulfur emissions was conducted to enable comparison with background conditions. Natural and anthropogenic emissions of chemical species were treated as described by Sheng et al. (2015). Each simulation was performed for 20 years representing atmospheric conditions of the years 2030–2049 for ozone depleting substances (2.3 ppb $Cl_Y$ and 18 ppt $Br_Y$
above 50 km altitude, WMO, (2008)) and GHG concentrations following the representative concentration pathway 6.0 scenario (RCP6.0), with the first 10 years used as spin-up and the last 10 years used for analysis. Sea surface temperatures (SST) and sea ice coverage (SIC) were prescribed as a repetition of monthly means of the year 2001 from the Hadley Centre Sea Ice and Sea Surface Temperature 1 data set (hadISST1) by the UK Met Office Hadley Centre (Rayner et al., 2003). The QBO was taken into account by a linear relaxation of the simulated zonal winds in the equatorial stratosphere to observed
wind profiles over Singapore perpetually repeating the years 1999 and 2000. The geoengineering emissions were injected at 50 hPa altitude (≈ 20 km) except for runs number 19 (termed "GEO_AERO_25km_15") and 20 (termed

"GEO_SO2_25km_15") which emitted AM-H$_2$SO$_4$ and SO$_2$, respectively, at about 24 hPa altitude ($\approx$ 25 km) to investigate the sensitivity to the emission altitude.

AM-H$_2$SO$_4$ emissions were parameterized as a log-normal distribution with a dry mode radius ($r_m$) of 0.095 µm and a distribution width ($\sigma$) of 1.5. This is the resulting size distribution determined by Pierce et al. (2010) from a plume model, derived at the point when coagulation with the larger background sulfate particles became dominant over self-coagulation. We also performed one run (number 21, termed "GEO_AERO_radii_00") with a mean radius of 0.15 µm to investigate the sensitivity of aerosol burden and radiative forcing to the initial AM-H$_2$SO$_4$ size distribution.

Runs 1 to 6 injected AM-H$_2$SO$_4$ while runs 7 to 12 injected SO$_2$ at the equator (i.e. $\pm$3.75° N and S), from 5° N to 5° S, 10° N to 10° S, 15° N to 15° S, 20° N to 20° S and from 30° N to 30° S, each uniformly spread over all longitudes. Emission into the tropical and subtropical stratosphere was chosen to achieve global spreading via the Brewer-Dobson circulation, and various latitudinal spreads were chosen to investigate sensitivity of emitting partly into the stratospheric surf zone and not only into the tropical pipe. The stratospheric surf zone is the region outside the subtropical transport barrier where breaking of planetary waves leads to quasi horizontal mixing (McIntyre and Palmer, 1984; Polvani et al., 1995). We assumed emission continuous in time, and injected into one vertical model level and the indicated emission area for all the scenarios, except for scenarios 13 and 14 which emitted AM-H$_2$SO$_4$ and SO$_2$, respectively, in two pulses per year (January 1$^{st}$-2$^{nd}$ and July 1$^{st}$-2$^{nd}$ of every modelled year) between 10° N and 10° S. Run 15 and 16 are scenarios with emissions into a single equatorial grid box (3.75° x 3.75° in longitude and latitude), whereas all the other scenarios emitted equally at all longitudes around the globe. With these scenarios, we investigated differences between a point source emission such as emissions resulting from a tethered balloon, and equally spread emissions such as emissions from continuously flying planes or a dense grid of continuously operating balloons.

We assume that AM-H$_2$SO$_4$ is produced in situ in the plumes behind planes which generate SO$_3$ or H$_2$SO$_4$ from burning elemental sulfur. As a 100 % conversion rate is unlikely to be achieved (Smith et al., 2018), we also performed two runs emitting mixtures of SO$_2$ and AM-H$_2$SO$_4$ with only 30 % or 70 % in the form of AM-H$_2$SO$_4$ and the rest in the form of SO$_2$ (runs 17 and 18, respectively).

Finally, we note that a fully coupled ocean would be desirable to study impacts on tropospheric climate such as surface temperature change. For computational efficiency, we chose not to couple the deep ocean module of SOCOL-AER in the present study. Therefore, we focus on changes in stratospheric aerosol microphysics, chemistry, and changes in surface radiation. We also note particular sensitivities to which horizontal and vertical resolution may play an important role.

## 4. Results

### 4.1 The stratospheric sulfur cycle under SSG conditions

A simplified representation of the modelled stratospheric sulfur cycle is shown for background conditions and for two geoengineering scenarios in Figure 1. Under background conditions (black numbers in figure), the stratospheric sulfur
burden arises primarily from cross-tropopause net fluxes of $SO_2$ and $SO_2$ precursor species (OCS, DMS, $CS_2$ and $H_2S$) as well as primary tropospheric sulfate aerosols in the rising air masses in the tropics. OCS contributes to the $SO_2$ burden in the middle stratosphere, while the direct cross-tropopause flux of $SO_2$ influences mainly the lower stratosphere (Sheng et al., 2015). In the tropical stratosphere, the mean stratospheric winds disperse sulfur species readily in east-west directions, whereas meridional transport is determined by large-scale stirring and mixing through the edge of the tropical pipe at typical
latitudes of 15°-20° N and S (Plumb, 1996). Transport into higher latitudes, which occurs mainly via the Brewer−Dobson circulation (Brewer, 1949; Dobson, 1956), results in a stratospheric aerosol residence time (= stratospheric burden divided by net flux to the troposphere) of about 13 months for background conditions in SOCOL-AER (see Table 2). After decomposition of sulfate precursors to $SO_2$ and subsequent oxidation to $H_2SO_4$ vapor, sulfate aerosol particles are forming via bimolecular nucleation with $H_2O$ or grow through condensation onto already existing aerosol particles. Condensation is
proportional to the available surface area of pre-existing aerosols and nucleation is mainly a function of temperature and $H_2SO_4$ partial pressure. The resulting stratospheric aerosol burden differs in some details (~16 % larger total stratospheric aerosol mass) from the one simulated in Sheng et al. (2015) due to different temporal sampling of model output as well as subsequent model updates and development.

      The aerosol burden resulting from geoengineering AM-$H_2SO_4$ injection for scenario GEO_AERO_15 (blue) is 41.4
20   % larger than the aerosol burden resulting from the equivalent $SO_2$ injection for scenario GEO_SO2_15 (red). The chemical lifetime of $SO_2$ in the lower stratosphere varies between 40 to 47 days among our $SO_2$ emission scenarios (about 31 days for background conditions). The $SO_2$ injection from GEO_SO2_15 results in an averaged stratospheric $SO_2$ burden of 222.4 Gg S for steady state conditions, which is 12.8 % of the combined stratospheric $SO_2$ and aerosol burden. This is a significant fraction compared to the 0.6 % in AM-$H_2SO_4$ emission scenarios. Especially when considering that the sulfur becomes only
"useful" for SSG after transformation to sulfate aerosols. In $SO_2$ emission scenarios only 4.7 % of the total emitted $SO_2$ is transported back to the troposphere unprocessed via diffusion and mixing due to tropopause folds at the edges of the tropical pipe. The other 95.3 % of the annually emitted $SO_2$ is subsequently oxidized to $H_2SO_4$ of which only 1 % is decomposed to $SO_2$ again through photolysis, 39 % nucleates to form new particles and 60 % condenses onto already existing particles. Compared to background conditions, GEO_AERO_15 shows a shift in the processing from nucleation to condensation due
to the increased surface area availability. Both SSG scenarios show an increased OCS flux across the tropopause (about +7 % among all scenarios) when comparing to the background run. This could be an indicator for enhanced upward mass fluxes across the tropical tropopause under SSG conditions or for decreased horizontal mixing from the tropical pipe to higher latitudes due to higher temperatures in the lower stratosphere (see Section 4.3) and thus modification of the Brewer-Dobson

circulation like observed in Visioni et al. (2017). Table 2 lists averaged values of aerosol burden, short-wave radiative forcing, and other quantities for all scenarios modelled, while Table 3 provides the quantities normalized by the corresponding all sky short-wave radiative forcing.

The $SO_2$ injection case produces both more nucleation mode (< 0.01 µm radius) particles and more coarse mode (>1 µm radius) particles than the AM-$H_2SO_4$ case (Fig. 2). GEO_SO2_15 shows about 3 orders of magnitude higher number concentrations of large particles in the coarse mode compared to GEO_AERO_15. The concentration of tiny nucleation mode particles is about 2-3 orders of magnitudes higher compared to BACKGROUND. This is due to the large nucleation rate driven by $H_2SO_4$ gas formed from $SO_2$ oxidation in GEO_SO2_15. The increase in coarse mode particles is partly due to $H_2SO_4$ condensing onto existing aerosols, and partly due to increased continuous coagulation of freshly formed nucleation mode particles with larger particles. The larger concentration of coarse mode particles in $SO_2$ emission scenarios leads to increased aerosol sedimentation rates and thus to 25.8 % shorter stratospheric aerosol residence times compared to AM-$H_2SO_4$ emission scenarios. We also show the 5th moment of the aerosol size distribution (see Figure 2, c), which gives an estimate of the downward mass flux due to aerosol sedimentation. This shows that particles in the size range 0.4-1.5 µm are contributing the most to sedimentation in the GEO_SO2_15 scenario. We can explain the total difference of 29.3 % smaller stratospheric aerosol burden in GEO_SO2_15 compared to GEO_AERO_15 by the 4.7 % of emitted $SO_2$ that gets lost to the troposphere unprocessed and the 25.8 % smaller stratospheric aerosol residence time.

In the AM-$H_2SO_4$ case, the number concentration of nucleation mode particles decreases below background conditions due to the increased surface area available for condensation (see Figure 2) and increased coagulation of nucleation mode particles with accumulation mode particles. For particles larger than 10 nm radius, the simulated distribution in the tropics is similar to the injected aerosol distribution with a peak of about 100 particles per $cm^3$ at about 0.1 µm radius. In Figure 2, the size range between 0.12 µm and 0.40 µm is highlighted in green as the range in which the efficacy of backscattering solar radiation on sulfate aerosols is at least 70 % of its peak value at 0.3 µm (solid green line, Dykema et al., 2016). In the tropics, the mass fraction of particles in the 0.12 to 0.40 µm size range is 0.79 of the total tropical aerosol mass for GEO_AERO_15 and 0.60 for GEO_SO2_15 (see also Table 3). Coagulation and sedimentation during transport to higher latitudes reduces the overall particle concentration in higher latitudes (dashed curves in Fig. 2) while increasing the mean particle size. Subsequently, the peak at about 0.1 µm in the tropics in GEO_AERO_15 becomes less pronounced and shifts slightly towards larger particles, which is closer to the radius of maximal backscattering of solar radiation on sulfate aerosols. Among all scenarios, the mass fraction in the optimal size range between 0.12 µm and 0.40 µm radius increases with transport to higher latitudes (e.g. 0.86 for GEO_AERO_15 and 0.67 for GEO_SO2_15 between 40° N and 60° N), which results in larger radiative forcing efficiency per unit stratospheric aerosol burden (see Fig. 2 and Table 3). Overall, AM-$H_2SO_4$ emission scenarios result in a more favourable aerosol size distributions, with more particles in the optimal size range for backscattering solar radiation compared to $SO_2$ emission scenarios.

The more favourable size distribution in AM-H$_2$SO$_4$ emission scenarios is also illustrated by values of effective radius ($r_{eff}$) for different scenarios (Fig. 3 and Table 2). The effective radius is the ratio of the 3$^{rd}$ moment to the 2$^{nd}$ moment of the size distribution. The size range between 0.24 µm and 0.36 µm, which is the range at which mass specific up-scatter is at least 90% of its peak backscattering efficiency at 0.3 µm is marked stippled in Figure 3. In GEO_AERO_15 (Fig. 3, a) this

$r_{eff}$ size range is seen in the lower stratosphere where aerosol mass concentrations are largest. However, GEO_SO2_15 (Fig. 3, d) shows $r_{eff}$ larger than is optimal for backscattering (up to 0.40 µm) in parts of the lower stratosphere. Our values of $r_{eff}$ for SO$_2$ emission scenarios are somewhat larger than what was computed by Niemeier et al. (2011), who found effective radii of about 0.3 µm in the lower stratosphere when emitting 2 Mt S in form of SO$_2$ at 60 hPa and 0.35 µm when emitting at 30 hPa. This is due to the different setup of the two studies. Niemeier et al. (2011) emitted at only one equatorial model grid

box. In our model, emission at one grid box also result in smaller particles with an effective radius of 0.33 µm averaged between 15° N and 15° S at 50 hPa (see Section 4.2, *spatio-temporal spread of emissions*).

## 4.2 Sensitivity simulations

*Latitudinal spread of emissions:* Previous studies found that the latitude range of emissions was important in determining size distribution and aerosol burden, and thus the resulting short-wave radiative forcing. English et al. (2012) found a 60 %

larger aerosol burden when emitting AM-H$_2$SO$_4$ between 32° N and 32° S when compared to emitting between 4° N and 4° S. They suggested that this is partly due to reduced aerosol concentrations and thus less coagulation as a result of a more dilute aerosol plume. However, they simultaneously increased the emission altitude, and thus they also state that the increased aerosol burden could partly be due to the increased stratospheric aerosol residence time at higher emission altitude. In contrast, Niemeier and Timmreck (2015) who emitted only at one equatorial model grid box found small decreases in

aerosol burden (4.3 %) when emitting between 30° N and 30° S relative to emitting from 5° N to 5° S. They found greater coagulation with more diffuse emissions, and that emission into the stratospheric surf zone increased cross-tropopause transport of aerosols and SO$_2$, thus resulting in a reduced stratospheric aerosol burden compared to scenarios that emitted SO$_2$ only into the tropical pipe.

      We find a small reduction in aerosol burden (<10 %) and clear sky short-wave radiative forcing for both AM-H$_2$SO$_4$

and SO$_2$ emission scenarios with increased latitudinal spread of the emissions (see Fig. 4). We assume that increased loss of stratospheric aerosols through tropopause folds in the surf zone with broadly spread emissions (>15° N-15° S) is compensated by increased coagulation and sedimentation of aerosols in scenarios which emit only into the tropical pipe.

      *Sensitivity to emission altitude:* The stratospheric aerosol residence time when emitting at 24 hPa (~25 km) is increased 28.6 % and 44.3 % when emitting AM-H$_2$SO$_4$ and SO$_2$ respectively, relative to emitting at 50 hPa (~20 km).

Therefore, GEO_AERO_25km_15 and GEO_SO2_25km_15 result in stratospheric aerosol burden of 2761 and 2190 Gg S respectively. In GEO_SO2_25km_15 the loss of unprocessed SO$_2$ to the tropopause is reduced to 2.3 % which is due to the greater distance of emissions from the tropopause and higher OH concentrations with increasing altitude in the stratosphere.

However, because of warmer temperatures at higher altitudes in the stratosphere, the nucleation rate decreases and a larger fraction of the $H_2SO_4$ gas condenses onto already existing particles (12.4 % nucleation, 87.6 % condensation). Therefore, particles tend to grow to even larger sizes and more accumulate in the coarse mode. The mass fraction of particles between 0.12 µm and 0.40 µm is reduced to only 0.41 and the all sky short-wave radiative forcing is only increased by 27.1 % in magnitude compared to GEO_SO2_15, which is not proportional to the increase in stratospheric aerosol burden (44.3 %). For GEO_AERO_25km_15 the fraction between 0.12 µm and 0.40 µm is only reduced from 0.79 to 0.77. This results in an all sky short-wave radiative forcing increase of 23.1 % compared to GEO_AERO_15, which is close to the increase in stratospheric aerosol burden (28.6 %). The latitudinal and vertical distribution of the aerosol mass density and the larger aerosol sizes are also seen in values of $r_{eff}$ (Fig. 3c and e). When emitting at 25 km, $r_{eff}$ in the tropical lower stratosphere increases from 0.35 µm to 0.52 µm in the $SO_2$ emission case and from 0.23 µm to 0.39 µm in the AM-$H_2SO_4$ emission case. Thus, the stippled area in Figure 3c and e ($r_{eff}$ of 0.24 to 0.36 µm) is below and above the lower stratosphere where most aerosol mass is accumulated. Therefore, emitting at 25 km results in less short-wave radiative forcing per resulting stratospheric aerosol burden. Furthermore, aerosol density remains more concentrated in the tropics when emitting at 25 km (see also Fig. 7a and b), due to a less leaky tropical pipe at higher stratospheric altitudes. However, when looking at the resulting clear sky short-wave radiative forcing (Fig. 7c and d), the peak in the tropics is only slightly increased in the $SO_2$ emission case and is even lower than the equivalent emission scenario at 20 km in the AM-$H_2SO_4$ emission case.

*Sensitivity to the injection mode radius:* GEO_AERO_radii_15 released AM-$H_2SO_4$ with mean radii of 0.15 µm instead of 0.095 µm. This results in slightly fewer but larger particles in the emitted aerosol plume. The increased injection radius resulted in slightly reduced aerosol burden due to either different coagulation and condensation regimes or faster sedimentation of the slightly larger particles. However, due to the increase of particles in the optimal size range for backscattering solar radiation (+11.4 %), the all sky short-wave radiative forcing increased 19.7 % compared to GEO_AERO_15 (see Figure 4). When looking at Tables 2 and 3, there is only a small difference in $r_{eff}$ between the AM-$H_2SO_4$ emission scenarios with $r_m$ of 0.095 µm and 0.15 µm, indicating only minor dependence of small changes in $r_m$ on the resulting distribution of accumulation mode particles.

*Mixtures of AM–$H_2SO_4$ and $SO_2$:* We also performed calculations to explore the utility of emitting $SO_2$ and AM-$H_2SO_4$ together (see Fig. 5). Some studies have suggested that planes carrying elemental sulfur and burning it in situ to directly emit $H_2SO_4$-gas or AM-$H_2SO_4$ would be the most effective way to deliver sulfate to the stratosphere (Benduhn et al. 2016, Smith et al. 2018). Burning elemental sulfur would also reduce the freight to be transported to the stratosphere (i.e. 32 g mol$^{-1}$ for sulfur rather than 98 g mol$^{-1}$ for $H_2SO_4$ or 64 g mol$^{-1}$ for $SO_2$). However, 100 % conversion to $H_2SO_4$ is unlikely, with the remainder emitted as $SO_2$. Our results show that compared to the pure AM-$H_2SO_4$ emission scenario GEO_AERO_15 a 30 % portion of $SO_2$ in the emission mixture (i.e. GEO_70%AERO_15) leads only to a slight reduction in the resulting stratospheric aerosol burden (-6.4 %) as well as in the mass fraction in the optimal size range for backscattering solar radiation and short-wave radiative forcing (see Table 2 and 3). And that a 30 % portion of AM-$H_2SO_4$ in an emission

mixture dominated by $SO_2$ (i.e. GEO_30%AERO_15) can increase stratospheric aerosol burden and short-wave radiative forcing significantly with respect to a pure $SO_2$ emission scenario (GEO_SO2_15). As the decrease in efficiency with increasing $SO_2$ portion in the emission mixture is strongly non-linear in our simulations, we do not expect significant efficiency loss for small portions of $SO_2$ in the emissions after in situ burning of elemental sulfur in planes.

*Spatio-temporal spread of emissions:* Emitting AM-$H_2SO_4$ in two pulses per year as assumed in GEO_AERO_pulsed_10 decreases the stratospheric aerosol burden by 3.8 % compared to the continuous emission scenario, GEO_AERO_10. This is due to the ~91 times larger aerosol mass concentration in the emission plume compared to GEO_AERO_10 (emission during 4 days per year instead of 365), and thus more coagulation and faster sedimentation in the emission region. Similar, but smaller effects were observed for GEO_AERO_point_00, which emitted at one equatorial grid

box only (3.75° x 3.75°), and thus with 96 times increased mass concentration in the emission region compared to GEO_AERO_00 (emissions spread over 3.75° in longitude instead of 360°). This indicates fast zonal mixing which reduces the effect of the higher initial mass concentration compared to the pulsed emissions. Changes in clear sky radiative forcing are about proportional to the aerosol burden, resulting in slightly smaller values for the sensitivity runs. However, all sky short wave radiative forcing is slightly larger compared to continuous scenarios emitting at all longitudes. This indicates

aerosol size distributions with more particles in the optimal size range (see Table 3) and thus more effective backscattering of solar radiation. Due to the increased coagulation, the particles are slightly larger with slightly more particles in the optimal size range for backscattering solar radiation.

       However, looking at GEO_SO2_pulsed_10 and GEO_SO2_point_00, which are the equivalent $SO_2$ emission scenarios, the stratospheric aerosol burden increased 10.0 % and 2.4 %, respectively compared to the continuous emission

scenarios at all longitudes. This is mainly due to a reduction in the total globally averaged aerosol nucleation rate (24.1 % nucleation, 75.9 % condensation) and the subsequent reduction in total coagulation. However, locally for GEO_SO2_point_00 and spatially for GEO_SO2_pulsed_10 the nucleation rates are increased in the emission area due to the greatly increased $H_2SO_4$ concentration in the emission region in these scenarios. These two scenarios produce spatially/temporally large amounts of nucleation mode particles which then quickly coagulate to produce accumulation mode

particles. After dilution when nucleation is small again, the continuous flow of tiny, freshly nucleated particles is disrupted, and coagulation is reduced. Thus, the mean particle diameter is smaller and the stratospheric aerosol residence time is increased compared to the respective continuous emission scenarios. Due to the decrease in particle diameter and thus more particles in the optimal size range (see Table 3) the all sky short-wave radiative forcing increases disproportionally compared to the stratospheric aerosol burden. All sky short wave radiative forcing is increased by 19.4 % and 8.3 % in

GEO_SO2_pulsed_10 and GEO_SO2_point_00 compared to GEO_SO2_00 and GEO_SO2_00, respectively (see Table 2). This partially mimics the processes in an $SO_2$ plume emitted by an aircraft. Other models also found single grid box $SO_2$ emissions to result in more effective radiative forcing (Niemeier et al., 2011) or found no difference compared to emissions

at all longitudes (English et al., 2012). However, the different behavior between point/pulsed $SO_2$ and point/pulsed AM-$H_2SO_4$ emission scenarios, which may be similar to solid aerosol particles is novel and has never shown before.

### 4.3 Temperature, OH, $H_2O$ and methane at the tropical cold point tropopause

Increased aerosol burden in the stratosphere also leads to heating of the lower stratosphere, mainly due to absorption of
longwave radiation. Heating of the cold point tropopause results in temperature increases of 0.7 to 1.2 K among all scenarios, associated with a $H_2O$ entry value increase of 0.30 to 0.55 ppmv (i.e. 9-17 %, see Fig. 6 and Table 2). The increased stratospheric $H_2O$ volume mixing ratio results in an increase of OH volume mixing ratio ($H_2O+O(^1D) => 2$ OH), which additionally increases the $HO_X$-ozone depletion cycle.

Figure 6 (c and d) shows how $H_2O$ volume mixing ratios increase above 200 hPa. The increase is slightly larger for
the AM-$H_2SO_4$ scenarios due to the higher aerosol load and more pronounced heating of the lower stratosphere. However, when comparing the vertical OH profiles (Fig. 6, a and b), the difference in OH between the $SO_2$ and the AM-$H_2SO_4$ scenarios increases to about 4 % at about 50 hPa, which is caused by the depletion of OH due to $SO_2$ oxidation ($SO_2+OH=>SO_3 +HO_2$).

Kleinschmitt et al. (2018) applied a mean lifetime of 41 days for $SO_2$ to $H_2SO_4$ conversion in their study and found
a $SO_2$-to-$H_2SO_4$ conversion rate of 96 %. They expected a slightly lower concentration in oxidants and therefore a longer $SO_2$ lifetime as well as a lower $SO_2$-to-$H_2SO_4$ conversion rate when taking chemical interactions into account. However, they did not consider the increased stratospheric $H_2O$ volume mixing ratio under SSG conditions which causes an OH concentration increase of up to 9 % in our model, making stratospheric $SO_2$ lifetime shorter and the $SO_2$-to-$H_2SO_4$ conversion rate larger.

There are other side effects of SSG such as tropospheric methane lifetime increase. In our simulation, tropospheric methane mixing ratios remain largely unchanged as we use prescribed mixing ratio boundary conditions for methane at the ground as well as prescribed SST. However, the reaction of methane with OH is strongly temperature dependent. In our model we observe a tropospheric temperature decrease of up to 0.95 K which leads to an increase in methane lifetime of up to 2.3 %, while OH concentration is 1almost unchanged in our simulations. Therefore, our model shows a similar effect as in Visioni et al. (2017),
but much smaller, as we emitted only 1.83 Mt S per year and Visioni et al. (2017) emitted 5 Mt S per year. When we scale our results linearly to 5 Mt S per year, methane lifetime increases by up to 6.3 % depending on the scenario. This is still less then the 10 % found by Visioni et al. (2017), probably due to the constant SST in our model setup. When taking interactive SST into account, increased changes in temperature, tropospheric ozone and $O^1(D)$ chemistry as well as $H_2O$ concentrations could account for the remaining difference to Visioni et al. (2017). In our simulations, the lifetime of methane at 50 hPa in the lower
stratosphere decreases about 14 % in continuous AM-$H_2SO_4$ emission scenarios at all longitudes and about 10 % in the corresponding $SO_2$ emission scenarios, which is in agreement with the OH changes described above.

## 4.4 AM–$H_2SO_4$ vs. $SO_2$ emissions

All sky short-wave radiative forcing from AM-$H_2SO_4$ emission scenarios 1-6 result in an average of 1.31 W m$^{-2}$. This value is 36.5 % larger compared to the average of the equivalent $SO_2$ emission scenarios 7-12, which result in 0.96 W m$^{-2}$. For clear sky short-wave radiative forcing the AM-$H_2SO_4$ scenarios 1-6 are on average 50 % then $SO_2$ emission scenarios 7-12.

Table 2 summarizes globally averaged quantities of all modelled SSG scenarios. The fifth column shows the ratios between surface clear sky and surface all sky short-wave radiative forcing. These values for the AM-$H_2SO_4$ emission scenarios are about 10 % larger than for the $SO_2$ emission scenarios, indicating a larger reduction of short-wave radiative forcing due to clouds and/or chemical interactions among AM-$H_2SO_4$ scenarios. The stratospheric aerosol burdens for the AM-$H_2SO_4$ scenarios are more concentrated within the tropical pipe between 15° N and 15° S compared to the equivalent $SO_2$ emission

scenarios that are more evenly distributed across all latitudes, as shown in Figure 7a and b. When looking at the clear sky short-wave radiative forcing (Fig. 7 c and d), the tropical peak flattens out compared to the peak in aerosol burden (Fig. 7 a and b) because the up-scattered fraction of solar radiation increases with increasing solar zenith angle. Additionally, in higher latitudes the aerosol size distributions show more particles in the optimal size range (green ranges in Fig. 2) for backscattering solar radiation, which also leads to increased backscatter efficiency per unit stratospheric aerosol burden. For

the $SO_2$ emission scenarios, the clear sky short-wave radiative forcing is almost equally distributed between 60° N and 60° S, whereas for the AM-$H_2SO_4$ scenarios, a pronounced peak can still be observed in the tropics where cloud cover is larger on average. Thus, the higher aerosol mass fraction in the cloudy tropics makes SSG less efficient in these regions. We assume the difference in surface clear sky to surface all sky short-wave radiative forcing ratios to be a result of a more favourable global spread of the resulting stratospheric aerosol burden for $SO_2$ emission scenarios. However, this indicates that emitting

only within the tropical pipe region might not be the optimal setup of SSG studies. Emitting at the edges of the tropical pipe at 15° N and 15° S as investigated by Tilmes et al. (2017) might be a more efficient way to achieve higher short-wave radiative forcing.

When looking at the depletion of the total ozone column (Fig. 7, e and f), we find larger depletion among the AM-$H_2SO_4$ emission scenarios. This is mainly due to the larger surface area densities for AM-$H_2SO_4$ scenarios in the emission

region in the tropics where ozone is produced. This leads to larger ozone depletion and therefore less ozone transport to higher latitudes. Therefore, among the AM-$H_2SO_4$ emissions scenarios, the vertical ozone column is depleted up to 7.5 % (for GEO_AERO_15 and GEO_AERO_00) at the south polar region in a 10-years average, whereas for the $SO_2$ emission scenarios, it is only up to 6 % (for GEO_SO2_10). The ozone depletion arises through formation of the reservoir species $HNO_3$ through $N_2O_5$ hydrolysis on aerosol surfaces which indirectly enhances the $ClO_x$ ozone depletion cycle. Furthermore,

chlorine gets activated through the heterogenous reaction of $ClONO_2$ with HCl, which contributes the most to the ozone depletion due to SSG.

Table 3 shows values from Table 2 normalized to the globally averaged all sky short-wave radiative forcing. The smallest absolute values are marked in green and the largest are marked in red. The smaller the absolute values, the larger the

short-wave radiative forcing, and/or the smaller the negative side effects investigated in this study. Other potential negative side effects like for example tropospheric cloud feedbacks (Visioni et al., 2018b) were not investigated in this study. Green values are accumulated among the AM-$H_2SO_4$ emission scenarios and red values are accumulated among the $SO_2$ emission scenarios. In average water vapor increase at the tropical cold point tropopause and global depletion of the ozone column are 15.5 % and 55.3 % larger, respectively, for AM-$H_2SO_4$ emission scenarios 1-6than for $SO_2$ emission scenarios 7-12 when normalizing to the emission rate of 1.83 Mt S yr$^{-1}$. However, when normalized by the resulting all sky short-wave radiative forcing, the increase in efficiency among the AM-$H_2SO_4$ emission scenarios outweighs the worse side effects in the AM-$H_2SO_4$ scenarios.

## 5. Discussion

In this study, the three-dimensional global aerosol-chemistry-climate model SOCOL-AER was used to investigate AM-$H_2SO_4$ and $SO_2$ emission scenarios for the purpose of SSG. We analysed the stratospheric sulfur cycle, aerosol burden, short-wave radiative forcing, and stratospheric temperature-$H_2O$-OH interactions for various SSG scenarios with injection at about 50 hPa ($\approx$ 20 km) altitude and two with injection at about 20 hPa ($\approx$ 25 km) altitude with a sulfur mass equivalent emission rate of 1.83 Mt S yr$^{-1}$ within the tropics and subtropics.

Direct continuous emission of aerosol particles in the accumulation mode at all longitudes and at 20 km altitude results in 37.8-41.4 % and 17.0-69.9 % larger stratospheric aerosol burden and all sky short-wave radiative forcing, respectively, compared to sulfur mass equivalent $SO_2$ emission scenarios. The difference in stratospheric aerosol burden is mainly because of two reasons. (1) AM-$H_2SO_4$ emissions have the advantage of demonstrating effects immediately after emission and not only after more than one month of transport, photochemistry and aerosol formation like in the case of $SO_2$ emissions (lifetime of $SO_2$ of 40 to 47 days through oxidation). Thus, direct AM-$H_2SO_4$ emission can create an immediate, targeted effect over the area of emission, whereas in $SO_2$ emission scenarios 12.8 % of the annually emitted sulfur is present in form of $SO_2$ on average. (2) The size distribution of $SO_2$ emission scenarios shows coarse mode particle concentrations which are about three orders of magnitudes larger compared to AM-$H_2SO_4$ emission scenarios. These particles sediment faster and reduce the average stratospheric residence time of the aerosols compared to AM-$H_2SO_4$ scenarios. In addition, the radiative forcing in $SO_2$ emission scenarios is influenced by the smaller mass fraction of particles in the optimal size range for backscattering solar radiation (i.e. 0.3 µm radius). The unfavourable size distribution for sulfate aerosols resulting from $SO_2$ emissions is largely due to condensation onto existing particles and the pronounced formation of tiny nucleation mode particles which subsequent coagulation with larger aerosols to create coarse mode particles.

Stratospheric aerosol burden and short-wave radiative forcing are about 10 % higher for scenarios which avoid emitting into the stratospheric surf zone, i.e. outside 15° N and 15° S. Enhanced loss of sulfur across the tropopause in scenarios emitting outside the tropical pipe is almost compensated by increased coagulation and thus sedimentation in scenarios which emit only into the tropical pipe. AM-$H_2SO_4$ emission scenarios additionally show a higher stratospheric

aerosol mass fraction in the tropics. This reduces all sky short-wave radiative forcing efficiency compared to $SO_2$ emission scenarios due to the higher cloud fraction in the tropics. Aerosol burden resulting from $SO_2$ emission scenarios are more equally spread to higher latitudes where an increased up-scatter fraction and a slightly better aerosol size distribution results in larger radiative forcing efficiencies per stratospheric aerosol burden. $AM-H_2SO_4$ emission scenarios result in slightly more stratospheric ozone depletion and stratospheric warming. However, due to the larger absolute short-wave radiative forcing, the negative side effects investigated in this study (i.e. stratospheric ozone and methane depletion, stratospheric temperature and $H_2O$ increase) are smaller for $AM-H_2SO_4$ emission scenarios compared to $SO_2$ emission scenarios when normalizing to the surface all sky short-wave radiative forcing.

On the one hand, for $AM-H_2SO_4$ emission scenarios, temporally and spatially increasing the mass density in the emission region leads to slightly shorter aerosol residence times through more coagulation. However, all sky short wave radiative forcing is slightly increased due to slightly more particles in the optimal size range for backscattering solar radiation. On the other hand, for $SO_2$ emission scenarios, a larger stratospheric aerosol burden for a fixed sulfur emission rate can be achieved by temporally or spatially increasing the $SO_2$ mass density. This strategy increases nucleation and coagulation rates in the emission region while minimizing nucleation and coagulation on a global scale, as first shown in Niemeier et al. (2011). The optimal frequency of the pulses as well as optimal spatial extent of the emissions requires further investigation. However, our results show different behavior of $AM-H_2SO_4$ and $SO_2$ emission scenarios to temporal and spatial spreads of the emissions. They also hint at a possible dependence on small-scale processes such as locally changing $SO_2$ and OH mass concentrations, which cannot be resolved in GCMs. This can be important when injecting emissions along the trajectory of an aircraft. Furthermore, the results underline the importance of interactions between chemistry and aerosols when modelling SSG scenarios.

We found that in the lower stratosphere OH concentrations are increased up to 8.8 % for $AM-H_2SO_4$ emission scenarios compared to the background run due to increased temperatures at the tropical cold point tropopause and thus higher water volume mixing ratios in the stratosphere. However, due to oxidation of $SO_2$ in $SO_2$ emission scenarios, the OH concentration increase is reduced to 4-5 % in the lower stratosphere compared to the background run in these scenarios.

We also examined scenarios in which mixtures of $SO_2$ and $AM-H_2SO_4$ were emitted. Pure $AM-H_2SO_4$ emission scenarios resulted in the largest stratospheric aerosol burdens as well as the largest clear sky and all sky short-wave radiative forcing. While the short-wave radiative forcing decreases with increasing fractions of $SO_2$, that increase is non-linear. A small fraction of $SO_2$ within emissions of $AM-H_2SO_4$ results in only slightly smaller radiative forcing efficiency, whereas a small fraction of $AM-H_2SO_4$ within emissions of $SO_2$ increases radiative forcing efficiency significantly. In situ burning of elemental sulfur in planes with subsequent conversion to $SO_3$ and, in the plume to $AM-H_2SO_4$ particles might therefore be effective in controlling the aerosol size distribution, even if conversion efficiency were significantly less than unity.

We found clear sky short-wave radiative forcing efficiencies of -1.22 W m$^{-2}$ and -0.82 W m$^{-2}$ per emitted Mt of sulfur equivalent injection rate (W m$^{-2}$ (Mt S yr$^{-1}$)$^{-1}$) for GEO_AERO_point_00 and GEO_SO2_point_00, respectively. For

all sky short-wave radiative forcing, the efficiencies were -0.77 W m$^{-2}$ and -0.57 W m$^{-2}$ per emitted Mt of sulfur, respectively. These values for point emission scenarios are somewhat smaller compared to other models such as for MAECHAM5 in Niemeier et al. (2011) and LMDZ-S3A in Kleinschmitt et al. (2018) who both got -0.60 W m$^{-2}$ (Mt S yr$^{-1}$)$^{-1}$ when emitting $SO_2$ into one model grid box at 19 km altitude and 17 km altitude, respectively. This is likely due to

5    differences in transport processes as well as lower stratospheric aerosol burdens in our model. On the one hand, this could be partially a result of different sedimentation, coagulation and aerosol binning schemes compared to other aerosol modules. On the other hand, SOCOL-AER slightly overestimates the Brewer–Dobson circulation (Dietmüller et al., 2018) when considering reference scenarios from the chemistry climate model initiative (CCMI). Even though stratospheric heating by SSG is not considered there, this could result in larger aerosol burdens compared to other models. When emitting $SO_2$ at 25

10    km altitude the efficiency is -0.67 W m$^{-2}$ (Mt S yr$^{-1}$)$^{-1}$ in SOCOL-AER (emissions at all longitudes) and -0.80 W m$^{-2}$ (Mt S yr$^{-1}$)$^{-1}$ for MAECHAM5 (Niemeier et al., 2011, with emissions at one grid box at 24 km). This is not proportional to the increase in stratospheric aerosol burden which is due to less favourable size distributions for backscattering solar radiation in these scenarios (see Section 4.2, *sensitivity to emission altitude*). However, as many processes are non-linear with increasing injection rates and altitudes, the efficiencies might be different for other SSG setups. Due to the lack of atmospheric

observations or small-scale field experiments, modelling studies are currently the only method to estimate efficiency and the possible adverse effects of SSG. Therefore, model intercomparison studies should further identify strengths and weaknesses among different models to reduce uncertainty. Furthermore, atmospheric field studies such as the Stratospheric Controlled Perturbation Experiment (SCoPEx, Dykema et al., 2014) could give further insight into stratospheric aerosol formation and plume evolution.

Additional uncertainties arise from the rather low resolution applied in this study. In particular, an increase in the vertical resolution as well as treatment of an interactive QBO could further increase the explanatory power of SSG studies with SOCOL-AER. To study tropospheric climate change, ocean feedback would have to be taken into account with the deep ocean module of SOCOL-AER. Furthermore, SOCOL-AER does not treat cloud interactions—which is likely one of the major uncertainties of the model as aerosols may have large impacts on clouds (Kuebbeler et al., 2012; Visioni et al.,

2018b). We only performed scenarios with emission regions limited to the tropics and subtropics centered around the equator. Nevertheless, SOCOL-AER is one of the first models that interactively couples a sectional aerosol module to the well-described chemistry and radiation schemes of a CCM.

      This study shows that direct emission of aerosols can give better control of the resulting size distribution, which subsequently results in more effective radiative forcing. Therefore, the SSG modelling community should increase their

focus on direct AM-$H_2SO_4$ emission as well as on the emission of solid particles (Weisenstein et al., 2015), such as calcite particles (Keith et al., 2016). Further investigations using SOCOL-AER to understand adverse effects of SSG, such as a closer look on ozone depletion, impacts on precipitation patterns, or on stratospheric dynamics will be conducted in future

studies. These studies will also include coupling of the deep ocean module of SOCOL-AER to investigate impacts on the tropospheric climate.

Furthermore, we show that interactive coupling of aerosols, chemistry and radiation schemes are essential features for modelling SSG emission scenarios with GCMs. For $SO_2$ emission scenarios, local depletion of oxidants was found, particularly with large $SO_2$ mass concentrations in the emission region. Accurate modelling of these scenarios might require higher temporal and spatial resolution than has been achieved with current GCMs. Therefore, coupling of small-scale Lagrangian plume dispersion models which simulate the first few days of aerosol-chemistry interactions and aerosol microphysics in evolving emission plumes from airplanes might be a desirable tool to improve future SSG modelling studies. This would appropriately account for the problem of connecting small scale temporal and spatial processes—such as aerosol formation, growth and evolution in an aircraft wakes—to the larger grid of GCMs, which has been neglected in past SSG studies.

## Author Contributions

The project was initialised by TP who had the lead together with DK. AS had the oversight and coordinated the whole study. AS and DW provided close supervision of SV during the whole project. SV adapted the SOCOL-AER model code to geoengineering purposes, performed all model simulations and analysed the model results, while all authors contributed to data interpretation. DW installed SOCOL-AER on the Harvard cluster. The design of the experiments was elaborated by all the authors under the lead of DK. AF helped with the model development. SV prepared the first manuscript with contributions from all co-authors. The paper writing process was financed by the Peter- and the Keith-group in equal parts.

## Acknowledgments

We want to thank Dr. Joshua Klobas and Dr. Jian-Xiong Sheng for the valuable discussions on the SOCOL-AER model as well as Dr. John Dykema, Dr. Lee Miller and Dr. Pete Irvine for helping to elaborate the different emission scenarios. Furthermore, we also want to thank the Harvard Research Computing team for the help in getting SOCOL-AER running on the Harvard cluster. Special thanks go to all the co-authors for enabling SV this very exiting master thesis exchange project at Harvard University and for the excellent supervision.

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

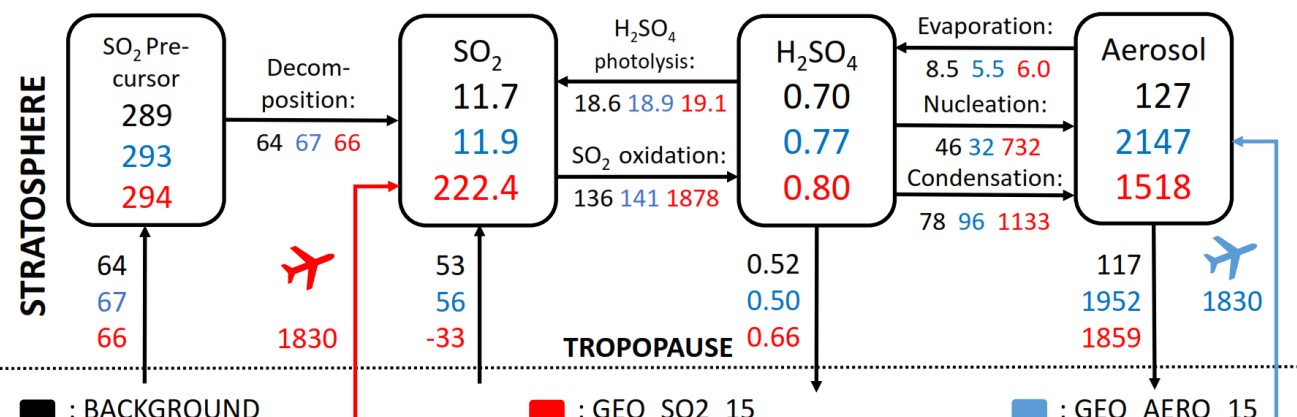

**Figure 1.** Schematic of the global sulfur cycle for GEO_AERO_15 (blue), GEO_SO2_15 (red), and BACKGROUND (black) averaged over the last ten years of simulations (see Table 1 for scenario definitions). Net fluxes (positive in pointing direction of arrow) are given in Gg S yr$^{-1}$ and burden (in boxes) are given in Gg S. SO$_2$ precursor species include OCS, DMS, H$_2$S, CS$_2$, and H$_2$S. SO$_3$ as an intermediate step between SO$_2$ and H$_2$SO$_4$ is modeled but for simplicity omitted from the diagram.

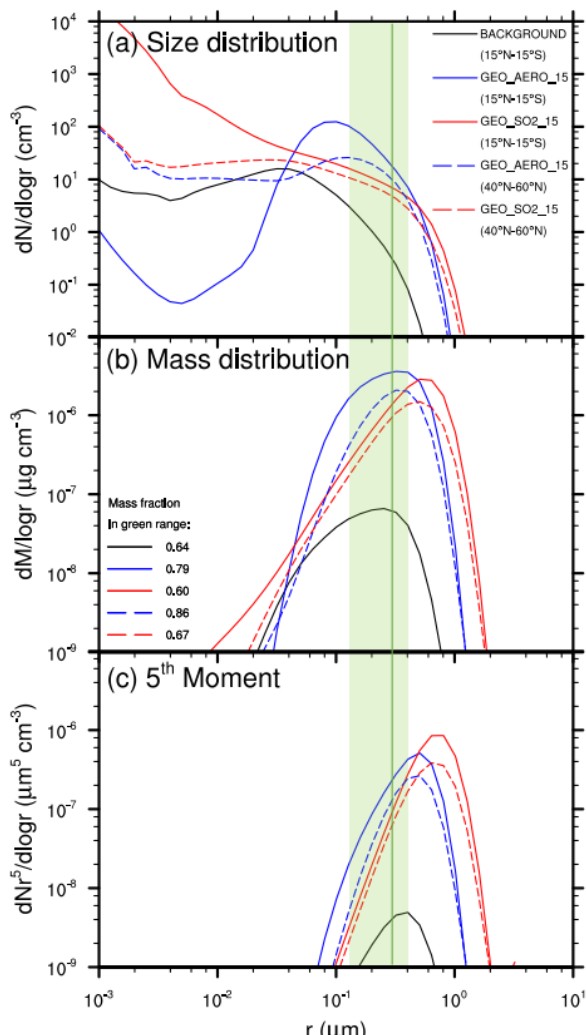

Figure 2. Size and mass distributions of stratospheric aerosol under various scenarios. (a) Wet aerosol size distributions of GEO_AERO_15 (blue), GEO_SO2_15 (red), and BACKGROUND (black) are zonally averaged over 10 years between 15° N and 15° S (continuous lines) and between 40° N and 60° N (dashed lines). Values shown are at 50 hPa in the tropics and at
5  100 hPa in the northern midlatitudes, i.e. at the levels of peak aerosol mass concentration in the vertical profile (see Fig. 3). The green size range is defined as the radius at which backscattering efficiency on sulfate aerosols is at least 70 % (i.e. 0.12-0.40 µm) of its maximal value (solid green line at 0.30 µm) following Dykema et al. (2016). (b) The resulting mass distributions of the size distribution curves shown in (a) including the wet aerosol mass fraction in the optimal size range between 0.12 and 0.40 µm in the legend. (c) The 5th moment of the aerosol size distribution shown in (a) as an estimate for
10  aerosol sedimentation mass flux.

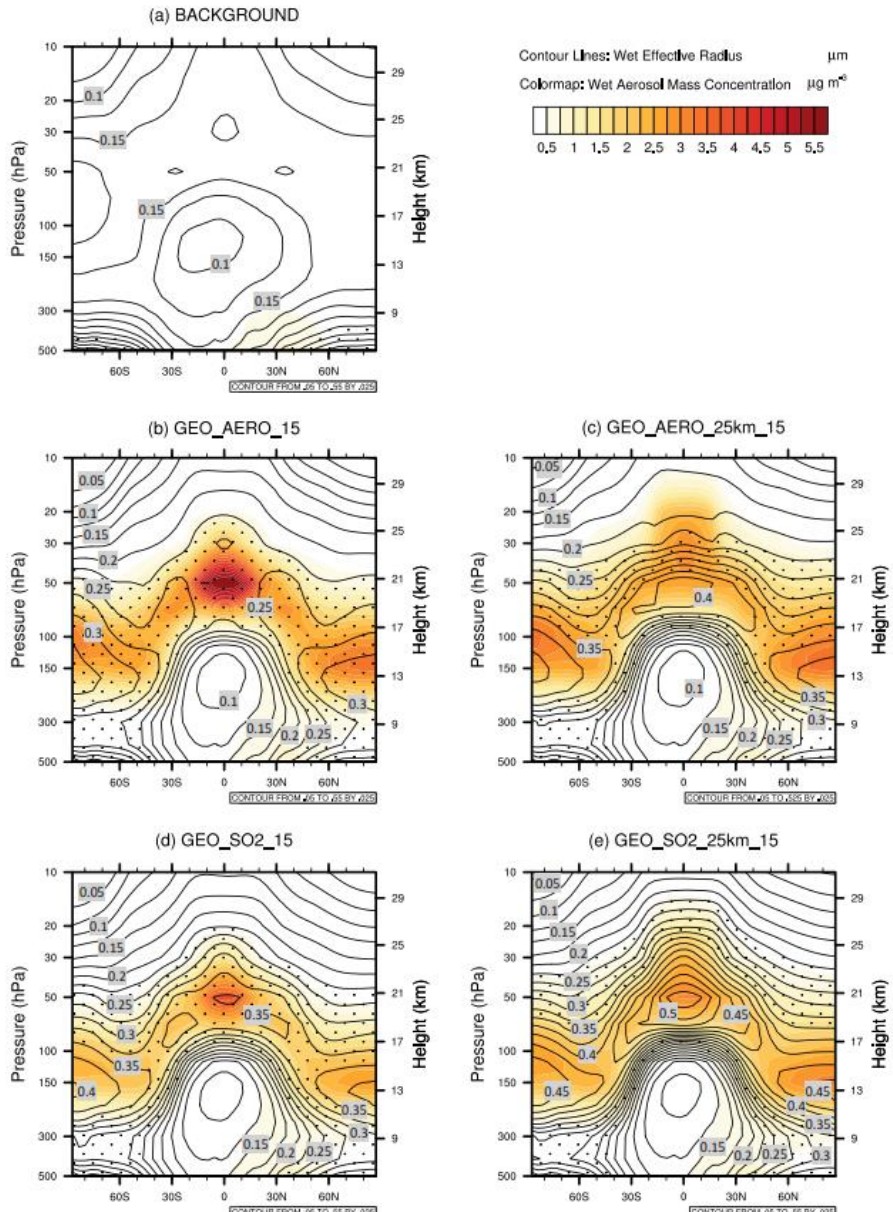

**Figure 3.** Contour lines show the wet effective radius (i.e. the ratio between the 3$^{rd}$ moment and the 2$^{nd}$ moment of the aerosol size distribution) in µm. The dotted area depicts the size range with effective radii between 0.24 µm and 0.36 µm. It is the range in which the backscattering efficiency is larger than 90 % of its peak value at 0.3 µm following Dykema et al. (2016). Color maps show the wet aerosol mass distribution in µg m$^{-3}$ both zonally averaged over ten years for background conditions **(a)**, GEO_AERO_15 **(b)**, GEO_AERO_25km_15 **(c)**, GEO_SO2_15 **(d)** and GEO_SO2_25km_15 **(e)**.

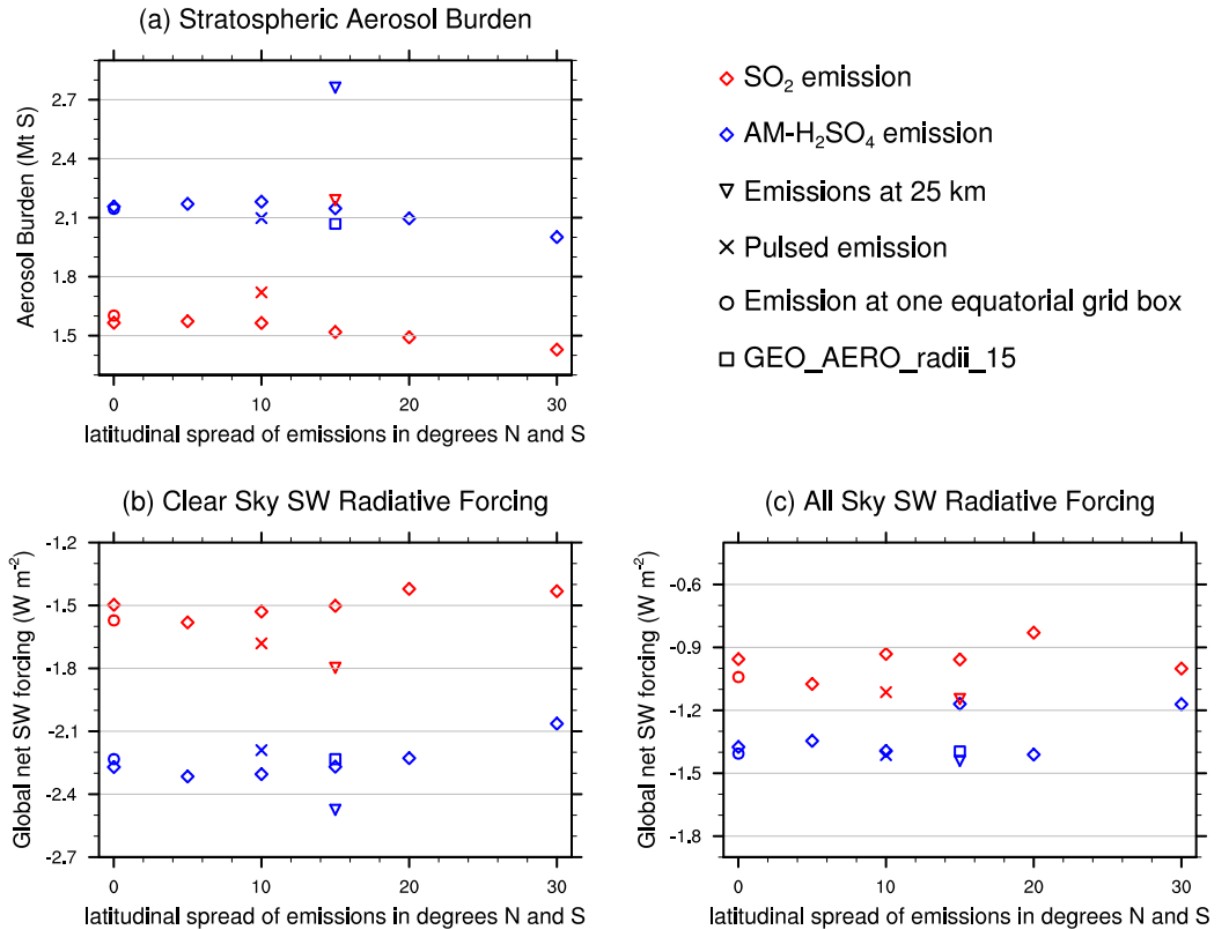

**Figure 4.** Globally averaged stratospheric aerosol burden **(a)**, clear sky **(b)**, and all sky **(c)** short-wave surface radiative forcing for various scenarios simulated in this study. AM-$H_2SO_4$ emission scenarios are shown in blue and $SO_2$ emission scenarios are shown in red. Reference scenarios are shown with "diamond" symbols. Alternate cases have symbols as indicated in key.

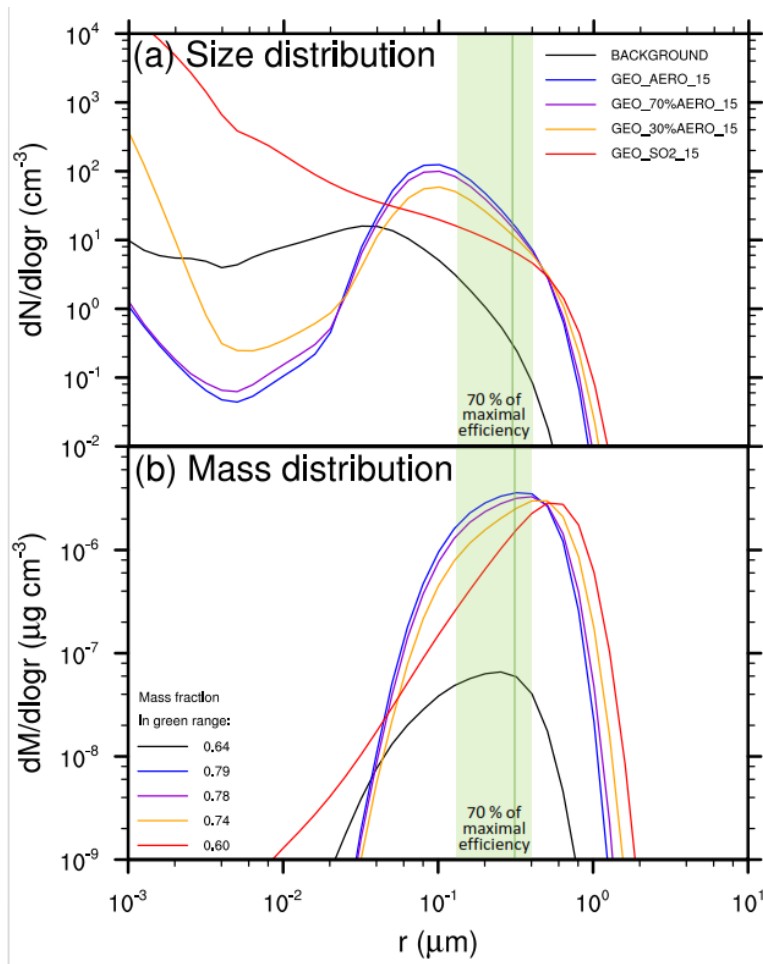

**Figure 5. (a)** Wet aerosol size distribution of scenarios with various emission mixtures of AM-H$_2$SO$_4$ and SO$_2$ averaged between 15° N and 15° S at 50 hPa. The green size range is defined as the radius at which backscattering efficiency on sulfate aerosols is larger than 70 % (i.e. 0.12–0.40 µm) of its maximal value (solid green line at 0.30 µm) following Dykema et al. (2016). **(b)** Mass distributions of the size distributions resulting from (a) including the wet aerosol mass fraction in the optimal size range between 0.12 and 0.40 µm in the legend.

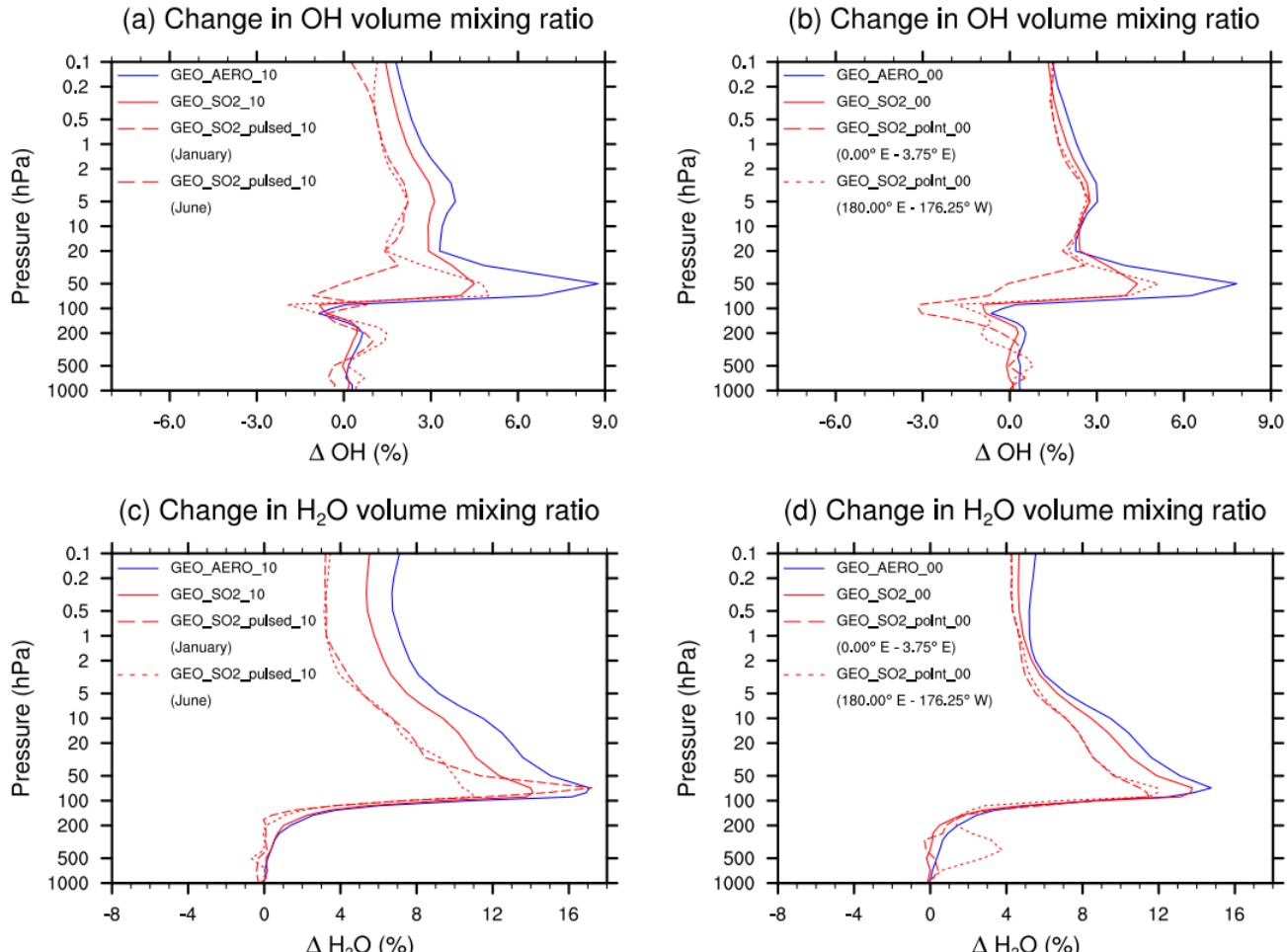

**Figure 6.** Vertical profile of the OH volume mixing ratio anomaly **(a and b)**, and $H_2O$ volume mixing ratio anomaly **(c and d)**. Anomalies represent prudential difference compared to the background simulation. Data show annual and zonal averages between 15° N and 15° S except where indicated. The left column **(a and c)** shows results for injections within 10° of the equator, while the right column **(b and d)** shows results for injections at the equator. For equatorial injection (right column), we show results averaged from 0° E to 3.75° E and from 180° E to 176.25° W. For the 10° injection case (left column), we show zonal averages of January (emissions during January 1st and 2nd) and June (month before emission) from the pulsed simulation. Note how $SO_2$ injection scenarios tend to reduce OH concentrations around 50 hPa compared with AM-$H_2SO_4$ injection scenarios (due to the reaction $SO_2+OH -> SO_3+HO_2$), while both increase stratospheric $H_2O$ concentrations due to warming.

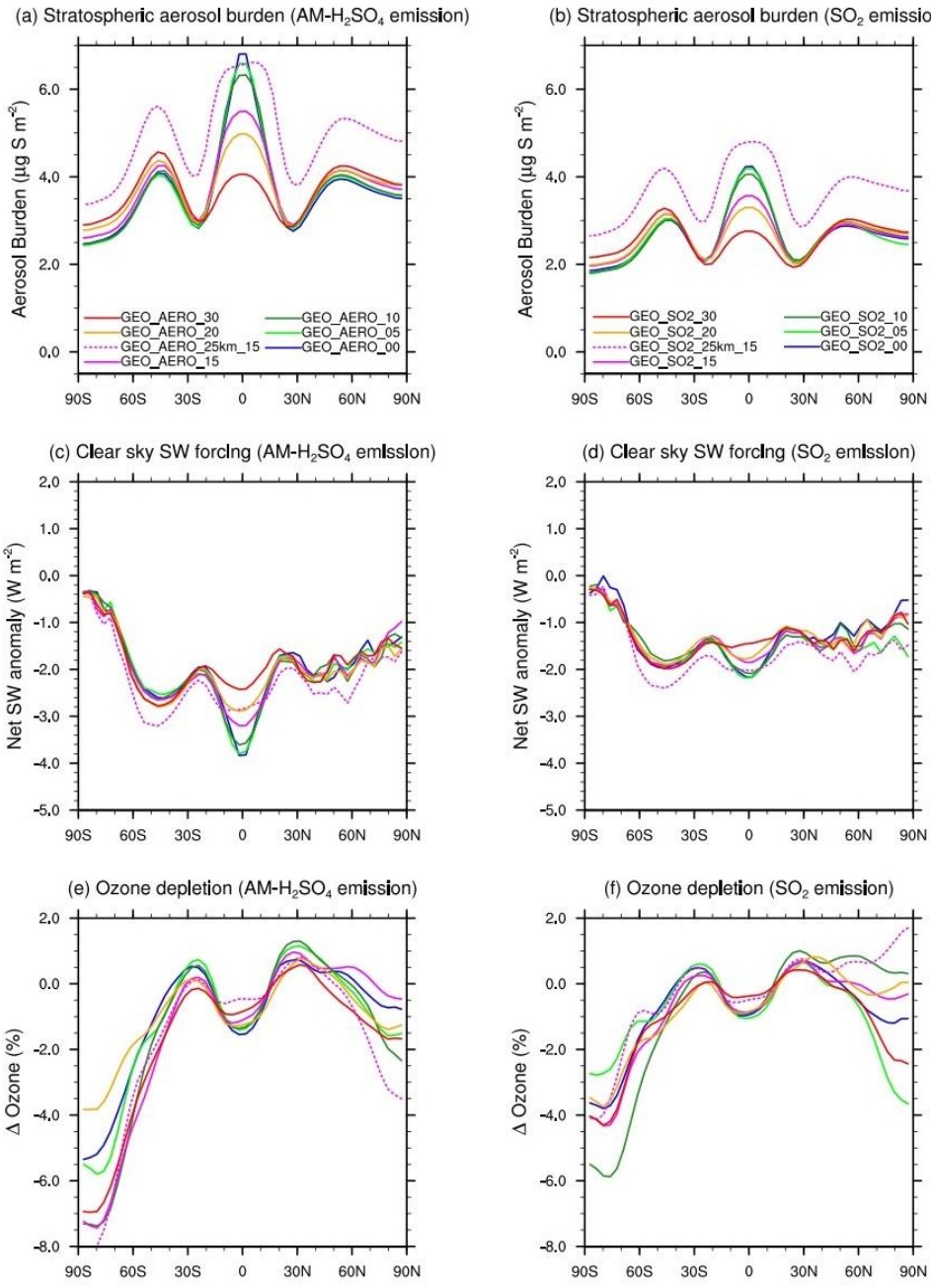

**Figure 7**. Zonally averaged stratospheric aerosol burden **(a and b)**, clear sky short-wave radiative forcing **(c and d)**, and depletion of the total ozone column **(e and f)** as a function of latitude for AM-H₂SO₄ **(a, c and e)** and SO₂ **(b, d and f)** emission scenarios with different latitudinal spread as well as for emissions at 25 km for comparison.

| Nr | Scenario | Longitude | Latitude | H$_2$SO$_4$ (liq) | SO$_2$ (gas) | Remarks |
|---|---|---|---|---|---|---|
| 0 | BACKGROUND | - | - | 0 % | 0 % | reference run |
| 1 | GEO_AERO_00 | 0°–360° E | 3.75° N–3.75° S | 100 % | 0 % | - |
| 2 | GEO_AERO_05 | 0°–360° E | 5° N–5° S | 100 % | 0 % | - |
| 3 | GEO_AERO_10 | 0°–360° E | 10° N–10° S | 100 % | 0 % | - |
| 4 | GEO_AERO_15 | 0°–360° E | 15° N–15° S | 100 % | 0 % | - |
| 5 | GEO_AERO_20 | 0°–360° E | 20° N–20° S | 100 % | 0 % | - |
| 6 | GEO_AERO_30 | 0°–360° E | 30° N–30° S | 100 % | 0 % | - |
| 7 | GEO_SO2_00 | 0°–360° E | 3.75° N–3.75° S | 0 % | 100 % | - |
| 8 | GEO_SO2_05 | 0°–360° E | 5° N–5° S | 0 % | 100 % | - |
| 9 | GEO_SO2_10 | 0°–360° E | 10° N–10° S | 0 % | 100 % | - |
| 10 | GEO_SO2_15 | 0°–360° E | 15° N–15° S | 0 % | 100 % | - |
| 11 | GEO_SO2_20 | 0°–360° E | 20° N–20° S | 0 % | 100 % | - |
| 12 | GEO_SO2_30 | 0°–360° E | 30° N–30° S | 0 % | 100 % | - |
| 13 | GEO_AERO_pulsed_10 | 0°–360° E | 10° N–10° S | 100 % | 0 % | two emission pulses per year (every 6 months) |
| 14 | GEO_SO2_pulsed_10 | 0°–360° E | 10° N–10° S | 0 % | 100 % | |
| 15 | GEO_AERO_point_00 | 0°–3.75° E | 3.75° N–3.75° S | 100 % | 0 % | emissions at only one equatorial grid box |
| 16 | GEO_SO2_point_00 | 0°–3.75° E | 3.75° N–3.75° S | 0 % | 100 % | |
| 17 | GEO_30%AERO_15 | 0°–360° E | 15° N–15° S | 30 % | 70 % | emission of different mixtures of SO$_2$ and AM–H$_2$SO$_4$ |
| 18 | GEO_70%AERO_15 | 0°–360° E | 15° N–15° S | 70 % | 30 % | |
| 19 | GEO_AERO_25km_15 | 0°–360° E | 15° N–15° S | 100 % | 0 % | emission altitude = 25 km |
| 20 | GEO_SO2_25km_15 | 0°–360° E | 15° N–15° S | 0 % | 100 % | |
| 21 | GEO_AERO_radii_15 | 0°–360° E | 15° N–15° S | 100 % | 0 % | $r_m$ = 0.15 μm |

**Table 1.** Overview of all simulations performed in this study. Each GEO scenario (1)-(12) assumes a zonally symmetric and continuous injection of 1.83 Mt S yr$^{-1}$ as SO$_2$ and/or accumulation mode particles (AM-H$_2$SO$_4$) with lognormal size distribution (dry mode radius, $r_m$ = 0.095 μm and distribution width $\sigma$ = 1.5). Longitudinal and latitudinal distributions as well as emission ratios of liquid to gas (in sulfur mass) are shown in columns 3-6. GEO scenarios (13)-(20) deviate from these standard conditions as indicated under remarks.

| Nr | Scenario | Aerosol Burden (Gg S) | Strat. Aerosol resid. time (months) | Clear Sky SW RF ($W\,m^{-2}$) | All Sky SW RF ($W\,m^{-2}$) | Clear Sky RF / All Sky RF | ΔOzone Column (%) | Tropo-pause ΔT (K) | $\Delta H_2O$ Volume Mixing Ratio (ppmv) | Effective Radius ($r_{eff}$) 40° N–60° N | Effective Radius ($r_{eff}$) 15° N–15° S |
|---|---|---|---|---|---|---|---|---|---|---|---|
| 0 | BACKGROUND | 127 | 13.03 | 0.00 | 0.00 | - | 0.00 | 0.00 | 0.00 | 0.13 | 0.14 |
| 1 | GEO_AERO_00 | 2156 | 13.25 | -2.27 | -1.38 | 1.65 | -0.49 | 0.95 | 0.40 | 0.27 | 0.24 |
| 2 | GEO_AERO_05 | 2169 | 13.33 | -2.32 | -1.35 | 1.72 | -0.49 | 1.15 | 0.52 | 0.27 | 0.23 |
| 3 | GEO_AERO_10 | 2181 | 13.39 | -2.30 | -1.39 | 1.65 | -0.65 | 1.17 | 0.51 | 0.27 | 0.23 |
| 4 | GEO_AERO_15 | 2147 | 13.20 | -2.27 | -1.17 | 1.94 | -0.75 | 1.22 | 0.55 | 0.27 | 0.23 |
| 5 | GEO_AERO_20 | 2096 | 12.87 | -2.23 | -1.41 | 1.58 | -0.57 | 0.95 | 0.41 | 0.26 | 0.23 |
| 6 | GEO_AERO_30 | 2002 | 12.30 | -2.06 | -1.17 | 1.76 | -0.98 | 0.84 | 0.37 | 0.24 | 0.22 |
| 7 | GEO_SO2_00 | 1565 | 10.36 | -1.50 | -0.96 | 1.57 | -0.36 | 0.96 | 0.42 | 0.33 | 0.35 |
| 8 | GEO_SO2_05 | 1571 | 10.30 | -1.58 | -1.07 | 1.47 | -0.43 | 1.04 | 0.46 | 0.33 | 0.35 |
| 9 | GEO_SO2_10 | 1564 | 10.16 | -1.53 | -0.93 | 1.64 | -0.36 | 0.99 | 0.44 | 0.33 | 0.35 |
| 10 | GEO_SO2_15 | 1518 | 9.80 | -1.50 | -0.96 | 1.57 | -0.43 | 0.88 | 0.38 | 0.33 | 0.35 |
| 11 | GEO_SO2_20 | 1491 | 9.61 | -1.42 | -0.83 | 1.71 | -0.44 | 0.82 | 0.33 | 0.33 | 0.34 |
| 12 | GEO_SO2_30 | 1428 | 9.12 | -1.43 | -1.00 | 1.43 | -0.51 | 0.84 | 0.36 | 0.32 | 0.34 |
| 13 | GEO_AERO_pulsed_10 | 2098 | 12.89 | -2.19 | -1.41 | 1.55 | -0.69 | 0.74 | 0.35 | 0.28 | 0.26 |
| 14 | GEO_SO2_pulsed_10 | 1720 | 11.62 | -1.68 | -1.11 | 1.51 | -0.41 | 0.84 | 0.31 | 0.30 | 0.30 |
| 15 | GEO_AERO_point_00 | 2145 | 13.18 | -2.23 | -1.41 | 1.59 | -0.76 | 0.87 | 0.42 | 0.28 | 0.25 |
| 16 | GEO_SO2_point_00 | 1602 | 14.24 | -1.57 | -1.04 | 1.51 | -0.10 | 1.03 | 0.34 | 0.31 | 0.33 |
| 17 | GEO_30%AERO_15 | 1829 | 11.55 | -1.84 | -1.11 | 1.66 | -0.75 | 0.97 | 0.38 | 0.30 | 0.28 |
| 18 | GEO_70%AERO_15 | 2010 | 12.52 | -2.09 | -1.14 | 1.82 | -0.87 | 0.80 | 0.44 | 0.27 | 0.24 |
| 19 | GEO_AERO_25km_15 | 2761 | 17.04 | -2.47 | -1.44 | 1.72 | -0.82 | 0.72 | 0.30 | 0.34 | 0.39 |
| 20 | GEO_SO2_25km_15 | 2190 | 13.80 | -1.80 | -1.22 | 1.48 | -0.26 | 0.75 | 0.33 | 0.41 | 0.52 |
| 21 | GEO_AERO_radii_15 | 2069 | 12.71 | -2.23 | -1.40 | 1.59 | -0.61 | 0.88 | 0.38 | 0.42 | 0.53 |

**Table 2.** Summarized values of all quantities calculated in this study. The first four columns show total stratospheric aerosol burden, globally averaged clear sky and all sky surface short-wave radiative forcing, as well as the ratio between the globally averaged clear sky and all sky short-wave radiative forcing. For ozone depletion, globally averaged values of the total ozone column reduction are shown in percentage points. Temperature increases and $H_2O$ volume mixing ratio increases are given as values averaged between 15° N and 15° S at 90 hPa (i.e. at the tropical cold point tropopause). The last 2 columns show wet effective radius ($r_{eff}$) averaged between 40° N and 60° N at 100 hPa and between 15° N and 15° S at 50 hPa.

| Nr | Scenario | S Injection Rate / ΔAll Sky SW RF | Aerosol Burden / ΔAll Sky SW RF | ΔOzone Column / ΔAll Sky SW RF | ΔH₂O / ΔAll Sky SW RF | ΔT / ΔAll Sky SW RF | Mass fraction 0.12—0.40 µm (40° N—60° N) | Mass fraction 0.12—0.40 µm (15° N—15° S) |
|---|---|---|---|---|---|---|---|---|
| 0 | BACKGROUND | - | - | - | - | - | 0.59 | 0.64 |
| 1 | GEO_AERO_00 | -1.32 | -1.57 | 0.36 | -0.29 | -0.69 | 0.87 | 0.80 |
| 2 | GEO_AERO_05 | -1.35 | -1.61 | 0.37 | -0.38 | -0.86 | 0.87 | 0.80 |
| 3 | GEO_AERO_10 | -1.31 | -1.56 | 0.47 | -0.37 | -0.84 | 0.87 | 0.79 |
| 4 | GEO_AERO_15 | -1.56 | -1.84 | 0.64 | -0.47 | -1.04 | 0.86 | 0.79 |
| 5 | GEO_AERO_20 | -1.30 | -1.49 | 0.40 | -0.29 | -0.67 | 0.86 | 0.80 |
| 6 | GEO_AERO_30 | -1.56 | -1.71 | 0.83 | -0.31 | -0.72 | 0.86 | 0.81 |
| 7 | GEO_SO2_00 | -1.90 | -1.64 | 0.38 | -0.44 | -1.01 | 0.68 | 0.60 |
| 8 | GEO_SO2_05 | -1.71 | -1.46 | 0.40 | -0.43 | -0.97 | 0.67 | 0.59 |
| 9 | GEO_SO2_10 | -1.96 | -1.68 | 0.39 | -0.47 | -1.07 | 0.66 | 0.59 |
| 10 | GEO_SO2_15 | -1.90 | -1.58 | 0.45 | -0.39 | -0.92 | 0.67 | 0.60 |
| 11 | GEO_SO2_20 | -2.20 | -1.80 | 0.53 | -0.40 | -0.98 | 0.67 | 0.61 |
| 12 | GEO_SO2_30 | -1.83 | -1.43 | 0.51 | -0.36 | -0.84 | 0.68 | 0.63 |
| 13 | GEO_AERO_pulsed_10 | -1.30 | -1.48 | 0.49 | -0.25 | -0.52 | 0.90 | 0.87 |
| 14 | GEO_SO2_pulsed_10 | -1.65 | -1.54 | 0.37 | -0.28 | -0.75 | 0.76 | 0.70 |
| 15 | GEO_AERO_point_00 | -1.30 | -1.52 | 0.54 | -0.30 | -0.62 | 0.89 | 0.85 |
| 16 | GEO_SO2_point_00 | -1.76 | -1.54 | 0.10 | -0.33 | -0.99 | 0.71 | 0.63 |
| 17 | GEO_30%AERO_15 | -1.65 | -1.37 | 0.67 | -0.34 | -0.88 | 0.79 | 0.74 |
| 18 | GEO_70%AERO_15 | -1.60 | -1.76 | 0.76 | -0.38 | -0.69 | 0.85 | 0.78 |
| 19 | GEO_AERO_25km_15 | -1.27 | -1.92 | 0.57 | -0.21 | -0.50 | 0.84 | 0.77 |
| 20 | GEO_SO2_25km_15 | -1.50 | -1.80 | 0.21 | -0.27 | -0.61 | 0.58 | 0.41 |
| 21 | GEO_AERO_radi_15 | -1.31 | -1.48 | 0.44 | -0.27 | -0.63 | 0.90 | 0.88 |

**Table 3.** Comparison of negative impacts investigated in this study normalized to the all sky short-wave radiative forcing. The smaller the absolute value of the ratio, the smaller the injection rate or impact to achieve a given level of radiative forcing. For each column, the smallest/largest three absolute values are marked in green/red. Columns show globally averaged values of sulfur injection rate, resulting global stratospheric aerosol burden and ozone depletion at 50 hPa as well as H₂O volume mixing ratio increase and temperature increases at the tropical cold point tropopause (i.e. at 90 hPa) normalized to the resulting all sky short-wave radiative forcing. Last two columns show the wet aerosol mass fraction in the range between 0.12 and 0.40 µm of the resulting size distribution. The three largest mass fractions are marked in green and the three smallest mass fractions are marked in red.