# Peer review of "Exploring accumulation-mode-H2SO4 versus SO2 stratospheric sulfate geoengineering in a sectional aerosol-chemistry-climate model"

_Atmospheric Chemistry and Physics, 2018_

## Short Comment (SC1) · 20 Nov 2018

I find this sentence in the abstract very confusing: "The modelled all-sky (clear-sky) shortwave radiative forcing for AM-H2SO4 injection scenarios is up to 17-70 % (44-57 %) larger than is the case for SO2." Since you use parentheses the normal way in the rest of the abstract, this usage makes the reader take a long time to figure out what you mean. Please just write this out as, "The modelled all-sky shortwave radiative forcing for AM-H2SO4 injection scenarios is up to 17-70% larger than is the case for SO2, and up to 44-57% larger for the clear-sky case." This takes little more space and is easy to

understand. I also have a question about the sentence. How can a value be up to a range of values? Shouldn't it be, "The modelled all-sky shortwave radiative forcing for AM-H2SO4 injection scenarios is up to 70% larger than is the case for SO2, and up to 57% larger for the clear-sky case." ?

I have not read the rest of the paper yet, but if you use parentheses like this in it, please change the text. See my article:

Robock, Alan, 2010: Parentheses are (are not) for references and clarification (saving space). Eos, 91(45), 419, doi:10.1029/2010EO450004. http://climate.envsci.rutgers.edu/robock/Parentheses2010EO450004.pdf

---

## Referee Comment (RC1) · Anonymous Referee #1 · 29 Nov 2018

Review of "*Exploring accumulation-mode-H2SO4 versus SO2 stratospheric sulfate geoengineering in a sectional aerosol-chemistry-climate model*" by S. Vattoni et al.

This article is an interesting comparison between different sulfate geoengineering (SG) injection strategies, namely between $SO_2$ injection (the standard for most simulations up to now) and $H_2SO_4$ (that has been proposed but not as often studied). The study is innovative in its comparison of the two injection strategies (not in the coupling of CCM and sectional aerosol model, as mentioned by the authors at the end of section 2), and I believe that it deserves publication in ACP after the two major issues (and several minor suggestions), that I list below, have been addressed.

Major points:

1) The authors compare their results with previously obtained results (some of them at least), but they have one major difference with them that is not highlighted anywhere: the longitudinal distribution of the emitted sulfate.
Let's take first Niemeier and Timmreck (2015) that the authors mention: they only have one simulation (Geo10-lon) where the injection is spread on all longitudes, and only for a very small latitudinal range (3N to Eq.). In all other scenarios, emissions are only between 120.9 to 123.75 E. Also Kleinschmitt et al. (2018): they mostly always inject at one single longitude (120E), and only once (BROAD) at 28 locations between 30N and 30S.
So, when the authors in the conclusion of their study claim that their RF efficiencies are smaller than those previous ones and they generally point at "lower stratospheric aerosol burdens in our model" they should explain this major difference.

I'm not suggesting the authors change their simulations to align closer to previous ones, but the difference between their study and previous ones should be further highlighted, and maybe compared with their sensitivity test (`GEO_AERO_point_00` and `GEO_SO2_point_00`) if they want to make an apple to apple comparison. Overall, I think it would be good that the authors justified much better their assumption that injections over all longitudes are preferable to injections centered in one longitude (and so they chose to do pointed injections only as sensitivity case). Do the authors think this is model dependant? For instance, Tilmes et al (2017) note that, in their model, " Single grid point injections produce sulfate aerosols of smaller size that reflect sunlight more efficiently than injections over a longitude band".

2) In Table 3 they show what they call "important impacts", naming water vapor and ozone column. However, recent literature has shown that those are not the only (nor the more important) impacts, especially in terms of radiative forcing: what about methane (Visioni et al., 2017 and Tilmes et al., 2018) or ice clouds (Kuebbeler et al., 2012 and Visioni et al, 2018a)? I believe that both factors can't be ignored, when talking about the possible range of impacts of SG. At the very least they should be mentioned, acknowledging previous results, but much better would be to show the changes produced in SOCOL regarding one or both of these aspects. Considering the authors focus also on chemical changes by showing OH and $H_2O$ vertical profiles, I suggest they at least analyse their atmospheric methane changes, comparing them with previous studies.

As I said, before being suitable for pubblications, I believe a major revision considering these two points is necessary. Furthermore, I have some minor points that I list below:

**Abstract:** The abstract is way too long and confusing. You don't really need to put all your results in the abstract, there's plenty of space elsewhere. Also, the use of the term "surprising" twice is missleading, considering that the last phrase of the abstract is "this study corroborates previous

studies". Shorten it by only pointing at some of the results and leave for the discussion all the rest. I agree with Alan Robock that the usage of parenthesis in that way needs to be dropped.

**P. 1, lines 17-19:** This phrase is very convoluted and confusing.

**P. 2, lines 15-25:** As I mentioned in my point 2), are the authors referring to only SG side-effects that reduce the efficiency of the RF produced by the sulfate aerosols (as stated in lines 17-18) or general drawback of SG? I believe the former would be more correct and would make more sense here. Because of this, a bit of clarification is important: (2) of all chemical side effects, ozone depletion is the one with the less significant RF effect (see Pitari et al., 2014). More important would be the effect on methane and other GHGs (see Visioni et al., 2017 and Tilmes et al., 2018) because of changes in photolysis rates and transport. Furthermore, in terms of RF, also the effects of a decrease in UT ice particles, as showed in Kuebbeler et al. (2012) and recently in Visioni et al (2018a), that would produce a cooling effect by trapping less planetary radiation, is important.

Point (4) has nothing to do with RF and possible effects of SG on ecosystem have been studied in more details after Kravitz et al. (2012) (such as in Xia et al., 2017), together with other changes (tropical storm ecc.). But this is not the point of this article.

**P. 4, line 8:** Visioni et al (2017, 2018a and 2018b) also used a sectional aerosol approach in their model and fully coupled microphysics and chemistry. It would be good to compare some of their results with yours since they also focus on RF changes, as I explain in point 2) and in some of these comments.

**P. 4, line 19:** Mills et al. (2017) is more of a description of WACCM-CESM new model set-up and the validation with Pinatubo data, so it is not about SG scenarios.

**P. 6, lines 11-13:** Here the authors should give the reader some more contest on this. The amount injected in this work is not really an issue here (because you are doing sensitivity studies and not looking at the long-term climatic response), but it's a bit of a stretch to say that that's how much Pinatubo injected. We don't know exactly how much $SO_2$ Pinatubo injected, so from a modeling prespective the amount of $SO_2$ for Pinatubo is very model dependant (from 10 to 20 $Tg$-$SO_2$ to get the best agreement with AOD observations), and it would be good to mention this (Timmreck et al. (2018) would be enough to reference, but just look at, for instance, Mills et al. (2016) where they inject 10 $Tg$-$SO_2$ and Pitari et al. (2016), where they inject 20 $Tg$-$SO_2$). Just give some contest to the reader. Personally, 1.83 Mt-S per year seems very arbitrary, and I would have preferred an amount more comparable to previous simulations.

**P. 7, line 2:** I would suggest explaining a bit what the authors mean by "surf zone" and offer some references (as you do, rightfully, for the BDC). It might not be such a common term as you think.

**P. 12, line 1:** As per my point 1): most other studies (see Tilmes et al., 2017, but also the entire GeoMIP G4 experiment) inject at only one longitude. In other simulations the single longitude was found either better for the scope ($r_{eff}$ closer to the desired one, Tilmes et al., 2017) or unimportant (because of the fast mixing time). You should mention this here.

**P. 12, line 5:** The shared longwave surface anomaly doesn't really say anything, especially since you don't have a surface coupling. It would not be due to the aerosol absorption anyway, but it is the result of many more processes. I suggest removing this phrase or explain better what you think the surface LW anomaly tells you, if you think it's important.

**P. 12, line 14:** constant climatological what? Concentration? It's confusing.

**P. 12, line 20:** split this in two phrases, one for AS and one for CS. Don't use parenthesis this way.

**P. 13, line 5:** Either you write "Emission strategies like that investigated by …" or "Emitting at … as investigated by …"

**P. 13, line 16-23:** Again: ozone and water vapor are only two of the SG side effects when it comes to RF: you should at least mention the fact that methane lifetime would increase and ice cloud decrease.

**P. 14, line 1:** like in the case of $SO_2$ emissions

**P. 15, lines 7-8:** as I said in my major point 1), mention the big difference between the studies, that is the different longitudinal distribution of the injections.

**P. 15, lines 13-14:** Yes, but in the framework of CCMI, so considering reference scenarios. The response to the BDC to the stratospheric heating produced by SG is not something that is considered there. The reference is still useful but mention this, at least.

**P. 15, line 24:** As has been previously studied in Kuebbeler et al (2012) and Visioni et al (2018a).

**P. 16, line 6:** "with" current GCMs.

**Figure 2:** This figure has a very poor resolution (compared to Fig. 5 that is similar). Panels b) and c) have such large scales and the curves are minuscule.

**Figure 4:** This is an interesting figure but must be improved. The scale for the SW forcing could be reduced a lot to higlight the differences between the different simulations (panel b from -1 to .3 W/m2 and panel c from 0 to -2 W/m2). Furthermore, symbols are very hard to read. I suggest enlarging them (or using more color).
Very minor thing, the **i**s and **l**s in figure 4 are all "weird", they look bold while the rest of the letters don't. You should fix this. (same goes for figure 2, 6 and 7). It's probably just a problem of how you saved the figures.

**References:**

Kleinschmitt, C., Boucher, O., and Platt, U.: Sensitivity of the radiative forcing by stratospheric sulfur geoengineering to the amount and strategy of the $SO_2$injection studied with the LMDZ-S3A model, Atmos. Chem. Phys., 18, 2769-2786, https://doi.org/10.5194/acp-18-2769-2018, 2018.

Kuebbeler, M., Lohmann, U., and Feichter, J.: Effects of stratospheric sulfate aerosol geo-engineering on cirrus clouds, Geophys. Res. Lett., 39, L23803, https://doi.org/10.1029/2012GL053797, 2012.

Niemeier, U. and Timmreck, C.: What is the limit of climate engineering by stratospheric injection of $SO_2$?, Atmos. Chem. Phys., 15, 9129-9141, https://doi.org/10.5194/acp-15-9129-2015, 2015.

Pitari, G., Di Genova, G., Mancini, E., Visioni, D., Gandolfi, I., and Cionni, I.: Stratospheric Aerosols from Major Vol- canic Eruptions: A Composition-Climate Model Study of the Aerosol Cloud Dispersal and e-folding Time, Atmosphere, 7, 79, https://doi.org/10.3390/atmos7060075, 2016.

Tilmes, S., J. H. Richter, M. J. Mills, B. Kravitz, D.G. MacMartin, F. Vitt, J. J. Tribbia, and J.-F. Lamarque, 2017: Sensitivity of aerosol distribution and climate response to stratospheric SO2 injection locations, JGR-Atmospheres

Tilmes, S., Richter, J. H., Mills, M. J., Kravitz, B., MacMartin, D. G., Garcia, R. R., et al. 2018: Effects of different stratospheric $SO_2$ injection altitudes on stratospheric chemistry and dynamics. *Journal of Geophysical Research: Atmospheres*, 123, 4654–4673

Timmreck, C., Mann, G. W., Aquila, V., Hommel, R., Lee, L. A., Schmidt, A., Brühl, C., Carn, S., Chin, M., Dhomse, S. S., Diehl, T., English, J. M., Mills, M. J., Neely, R., Sheng, J., Toohey, M., and Weisenstein, D.: The Interactive Stratospheric Aerosol Model Intercomparison Project (ISA-MIP): motivation and experimental design, Geosci. Model Dev., 11, 2581-2608, https://doi.org/10.5194/gmd-11-2581-2018, 2018.

Visioni, D., Pitari, G., Aquila, V., Tilmes, S., Cionni, I., Di Genova, G., and Mancini, E.: Sulfate geoengineering impact on methane transport and lifetime: results from the Geoengineering Model Intercomparison Project (GeoMIP), Atmos. Chem. Phys., 17, 11209-11226, https://doi.org/10.5194/acp-17-11209-2017, 2017.

Visioni, D., Pitari, G., di Genova, G., Tilmes, S., and Cionni, I.: Upper tropospheric ice sensitivity to sulfate geoengineering, Atmos. Chem. Phys., 18, 14867-14887, https://doi.org/10.5194/acp-18-14867-2018, 2018a.

Visioni, D., Pitari, G., Tuccella, P., and Curci, G.: Sulfur deposition changes under sulfate geoengineering conditions: quasi-biennial oscillation effects on the transport and lifetime of stratospheric aerosols, Atmos. Chem. Phys., 18, 2787-2808, https://doi.org/10.5194/acp-18-2787-2018, 2018b.

Xia, L., Nowack, P. J., Tilmes, S., and Robock, A.: Impacts of stratospheric sulfate geoengineering on tropospheric ozone, Atmos. Chem. Phys., 17, 11913-11928, https://doi.org/10.5194/acp-17-11913-2017, 2017.

---

## Referee Comment (RC2) · Anonymous Referee #2 · 12 Dec 2018

Manuscript by Vattioni et al studies several injection strategies for stratospheric sulfur geoengineering with gas phase SO2 and sulfate droplet injections. Research is done by using global 3D-aerosol-chemistry-climate model. This is one of the few studies where the impacts of stratospheric sulfur injections are studied with a sectional aerosol model coupled with/included in a global climate model. Authors have simulated several different scenarios to cover a wide range of options to inject sulfur to stratosphere.

Even though the general idea of the study and the studied scenarios are not totally new, the study shows several eye opening results and it is a good addition to existing

research. Currently there are relatively few studies where stratospheric sulfur geoengineering is simulated by including aerosol microphysics and especially with sectional aerosol model. Radiative forcing of stratospheric sulfur geoengineering is dependent on several factors, related to how sulfur is injected, but also how the microphysics is modelled. Thus it is valuable to get information from different scenarios simulated with different models. Authors also quantify microphysical processes (such as nucleation, condensation, coagulation) in various scenarios which helps to understand the impacts of microphysical processes on geoengineering. In addition, for example, the responses in OH concentration were surprising, but well justified. Overall this is an interesting and excellent study. It is well written and does not leave open questions. Thus I recommend publishing this manuscript and I have only minor comments on some specific points in the text. I also have to say that it is quite impressive that the work is based on a master's thesis.

I want to mention that I do not agree with reviewer 1 concern about differences between longitudinal distribution of emitted sulfur. As it is generally known, and pointed out in this study, results from point like simulations do not differ much from injections over all longitudes. In addition, it would be challenging to do an apple to apple comparison between the results of this and earlier studies, and I think it is not necessary in this case. My opinion is that the author's choice to use "all longitudes" -case as "default" option and pulsed scenario as a sensitivity case would have been natural choice for me too.

P1, l 17 As was already commented by Alan Robock, using parentheses like this is a bit confusing.

P1 , l22-23 " Increasing the local SO2 flux in the injection region by pulse or point emissions reduces the..." Would it be better to say something like: "concentrating injections to smaller regions by pulse or..." You don't just increase emissions somewhere but simultaneously decrease (remove) them elsewhere.

[Figure]

P2, l20 I would include following citation: Niemeier, U. and Timmreck, C.: What is the limit of climate engineering by stratospheric injection of SO2?, Atmos. Chem. Phys., 15, 9129-9141, https://doi.org/10.5194/acp-15-9129-2015, 2015. It shows nicely the reduced efficiency in the case of really high loading.

P2, l22 Is it really a limitation? This study is not concentrating on this topic so this sentence can be removed.

P4, L2 Just a comment, sigma is usually fixed and same mode width does not represent well both coarse mode particles in troposphere and stratosphere (long living particles).

P4, L8. Sectional aerosol model is also used in: Laakso, A., Kokkola, H., Partanen, A.-I., Niemeier, U., Timmreck, C., Lehtinen, K. E. J., Hakkarainen, H., and Korhonen, H.: Radiative and climate impacts of a large volcanic eruption during stratospheric sulfur geoengineering, Atmos. Chem. Phys., 16, 305-323, https://doi.org/10.5194/acp-16-305-2016, 2016.

Laakso, A., Korhonen, H., Romakkaniemi, S., and Kokkola, H.: Radiative and climate effects of stratospheric sulfur geoengineering using seasonally varying injection areas, Atmos. Chem. Phys., 17, 6957-6974, https://doi.org/10.5194/acp-17-6957-2017, 2017.

P4, L18. , "the radiation scheme did not interact with the aerosol module" This is not true.

P6, L11 Is there some explanation behind the decision to use 1.83 MT S yr-1 injections? For me it sounds like an accidental choice where you originally planned to do injections with certain mass but after all simulations were done, you noticed that unit in emission(/injection) was not what it should have been. However, I do not say that this is a problem, because there is not anything "wrong" to use this value, but if there is a sensible reason for use this specific value, it should be mentioned.

This is also just a comment, but it would have been nice to see differences between

SO2 and sulfate injections in a case of larger amount of injection.

P6, L13 There are several estimations for mass of the emitted sulfur from Mt Pinatubo eruption. It would be good to cite some study.

P6, L21 QBO nudging (without nudging winds generally) is new to me. If you can open this method by few clear sentences, it would be great. If not, then it is ok as it is.

P8, L28 and L33, Based on table 2, I got 26.8% shorter resid. time in GEO_SO2_15 than in corresponding AERO-case (not 23.3%). What is 32% difference in L30? It would not be the first time that I cannot calculate something right but please check these.

P9, L2 and maybe due to the coagulation?

P9-> It would be useful if radiative forcing for LW was mentioned at some point. Kleinschmitt et al. 2018 got quite large LW forcing values compared to other studies and it would be interesting to see how this is in the model used in this study. I expect that there is not much (absolute) difference between cases where sulfur is injected as SO2 or sulfate (?).

P10, L4. Reduction is seen only in clear-sky forcing but not in all-sky.

P10, L6 As was pointed out by reviewer 1 too, I had to google "surf zone" so maybe it is not that familiar word.

P10, L21. It is better to use 25km instead of 24hPa to be consistent with experiment names.

P11, L2 0.95 -> 0.095 um

P11, L11 compared to . . .

P12, L5 Based on my experience, aerosols are not affecting much on LW fluxes at the surface. This line ("The longwave surface. . .") can be removed.

P12, L14. "constant climatological SO2"? What does it mean?

P12, L21 Parenthesis thing - same as in abstract

P12, L29-30 Just a comment: I don't know has this been pointed out in some earlier studies, but if it has, at least I have missed it. This was an interesting remark and it sounds credible. In addition, the size distribution of particles is different in tropical peak compared to higher latitudes.

P13. L18 "...the smaller the negative side effects" Can you really say this? There are several negative side effects which are not studied here.

P14, L29 "are only increased by about 4%" I would remove word "only". I was surprised that OH concentration was generally increased.

Figures: In addition to reviewer 1 comments please correct following typos: Fig3: Areosol -> aerosol (in upper right) Fig6: Janauary -> January

---

## Author Comment (AC1) · 22 Feb 2019

Please find the response in the attached file.
* * *

---

## Author Comment (AC2) · 22 Feb 2019

**Review of "*Exploring accumulation-mode-H2SO4 versus SO2 stratospheric sulfate geoengineering in a sectional aerosol-chemistry-climate model*" by S. Vattioni et al.**

**Comments by anonymous reviewer #2 are in bold.** Author responses are in blue.

**Manuscript by Vattioni et al studies several injection strategies for stratospheric sulfur geoengineering with gas phase SO2 and sulfate droplet injections. Research is done by using global 3D-aerosol-chemistry-climate model. This is one of the few studies where the impacts of stratospheric sulfur injections are studied with a sectional aerosol model coupled with/included in a global climate model. Authors have simulated several different scenarios to cover a wide range of options to inject sulfur to stratosphere.**

**Even though the general idea of the study and the studied scenarios are not totally new, the study shows several eye-opening results and it is a good addition to existing research. Currently there are relatively few studies where stratospheric sulfur geoengineering is simulated by including aerosol microphysics and especially with sectional aerosol model. Radiative forcing of stratospheric sulfur geoengineering is dependent on several factors, related to how sulfur is injected, but also how the microphysics is modelled. Thus, it is valuable to get information from different scenarios simulated with different models. Authors also quantify microphysical processes (such as nucleation, condensation, coagulation) in various scenarios which helps to understand the impacts of microphysical processes on geoengineering. In addition, for example, the responses in OH concentration were surprising, but well justified. Overall this is an interesting and excellent study. It is well written and does not leave open questions. Thus, I recommend publishing this manuscript and I have only minor comments on some specific points in the text. I also have to say that it is quite impressive that the work is based on a master's thesis.**

**I want to mention that I do not agree with reviewer 1 concern about differences between longitudinal distribution of emitted sulfur. As it is generally known, and pointed out in this study, results from point like simulations do not differ much from injections over all longitudes. In addition, it would be challenging to do an apple to apple comparison between the results of this and earlier studies, and I think it is not necessary in this case. My opinion is that the author's choice to use "all longitudes" -case as "default" option and pulsed scenario as a sensitivity case would have been natural choice for me too.**

We would like to thank reviewer #2 for her/his insightful comments and suggestions. Please find our detailed replies below:

**P1, L17: As was already commented by Alan Robock, using parentheses like this is a bit confusing.**

Corrected.

**P1, L22-23: "Increasing the local SO2 flux in the injection region by pulse or point emissions reduces the. . ." Would it be better to say something like: "concentrating injections to smaller regions by pulse or. . ." You don't just increase emissions somewhere but simultaneously decrease (remove) them elsewhere.**

Was changed to: "In the case of SO$_2$ emissions, limiting the sulfur injections spatially and temporally in form of point and pulsed emissions reduces the total global annual nucleation… "

**P2, L20: I would include following citation: Niemeier, U. and Timmreck, C.: What is the limit of climate engineering by stratospheric injection of SO2?, Atmos. Chem. Phys., 15, 9129-9141, https://doi.org/10.5194/acp-15-9129-2015, 2015. It shows nicely the reduced efficiency in the case of really high loading.**

The citation was added.

**P2, L22: Is it (i.e. point 4) really a limitation? This study is not concentrating on this topic so this sentence can be removed.**

Our original "point (4)" has been removed as it is not a limitation in terms of radiative forcing.

**P4, L2 Just a comment, sigma is usually fixed and same mode width does not represent well both coarse mode particles in troposphere and stratosphere (long living particles).**

Corrected.

**P4, L8: Sectional aerosol model is also used in:**
**Laakso, A., Kokkola, H., Partanen, A.- I., Niemeier, U., Timmreck, C., Lehtinen, K. E. J., Hakkarainen, H., and Korhonen, H.: Radiative and climate impacts of a large volcanic eruption during stratospheric sulfur geoengineering, Atmos. Chem. Phys., 16, 305-323, https://doi.org/10.5194/acp-16- 305-2016, 2016.**
**Laakso, A., Korhonen, H., Romakkaniemi, S., and Kokkola, H.: Radiative and climate effects of stratospheric sulfur geoengineering using seasonally varying injection areas, Atmos. Chem. Phys., 17, 6957-6974, https://doi.org/10.5194/acp-17-6957-2017, 2017.**

The references to the two paper have been added, including a short discussion about their model.

**P4, L18: "the radiation scheme did not interact with the aerosol module" This is not true.**

This part of the sentence was removed.

**P6, L11: Is there some explanation behind the decision to use 1.83 MT S yr-1 injections? For me it sounds like an accidental choice where you originally planned to do injections with certain mass but after all simulations were done, you noticed that unit in emission(/injection) was not what it should have been. However, I do not say that this is a problem, because there is not anything "wrong" to use this value, but if there is a sensible reason for use this specific value, it should be mentioned. This is also just a comment, but it would have been nice to see differences between SO2 and sulfate injections in a case of larger amount of injection.**

The initial idea was to emit 2 Mt/yr, but due to a problem in the emission scheme, effectively only 1.83 Mt/yr have been emitted. As you already mentioned, this has no effect on the conclusions made in this paper as the same amount of sulfur has been emitted in all our model simulations. The 1.83 Mt/yr correspond to 9-20% of the emitted amount of sulfur from the Mt. Pinatubo eruption, depending which model one considers as the estimate of total S emission ranges from about 10-20 Mt S. Unfortunately, we did not have the resources to conduct simulations with different emission magnitudes. We clearly declare that the sensitivities with respect to aerosol burden and radiative forcing vary for different emission magnitudes.

**P6, L13: There are several estimations for mass of the emitted sulfur from Mt Pinatubo eruption. It would be good to cite some study.**

We used Sukhodolov et al. (2018) as a reference, who conducted a Pinatubo study with SOCOL-AER.

**P6, L21: QBO nudging (without nudging winds generally) is new to me. If you can open this method by few clear sentences, it would be great. If not, then it is ok as it is.**

I changed the sentence to: "The QBO was taken into account by a linear relaxation of the simulated zonal winds in the equatorial stratosphere to observed wind profiles over Singapore perpetually repeating the years 1999 and 2000."

**P8, L28 and L33: Based on table 2, I got 26.8% shorter resid. time in GEO_SO2_15 than in corresponding AERO-case (not 23.3%). What is 32% difference in L30? It would not be the first time that I cannot calculate something right but please check these.**

You are right. Thank you very much for spotting this mistake. All calculations in the paper have been recalculated and corrected if necessary.

**P9, L2: and maybe due to the coagulation?**

Corrected to: "In the AM-$H_2SO_4$ case, the number concentration of nucleation mode particles decreases below background conditions due to the increased surface area available for condensation (Fig. 2) and increased coagulation of nucleation mode particles with accumulation mode particles."

**P9: It would be useful if radiative forcing for LW was mentioned at some point. Kleinschmitt et al. 2018 got quite large LW forcing values compared to other studies and it would be interesting to see how this is in the model used in this study. I expect that there is not much (absolute) difference between cases where sulfur is injected as SO2 or sulfate (?).**

We did not look at LW radiation as we used constant sea surface temperatures. We only observed surface LW radiation anomalies of <0.1 $W/m^2$. For a solid LW anomaly estimate a coupled ocean module would be necessary.

**P10, L4: Reduction is seen only in clear-sky forcing but not in all-sky.**

Corrected.

**P10, L6: As was pointed out by reviewer 1 too, I had to google "surf zone" so maybe it is not that familiar word.**

Was corrected to "…emitting partly into the stratospheric surf zone and not only into the tropical pipe. The stratospheric surf zone is the region outside the subtropical transport barrier where breaking of planetary waves leads to quasi horizontal mixing (McIntyre and Palmer, 1984; Polvani et al., 1995)."

**P10, L21: It is better to use 25km instead of 24hPa to be consistent with experiment names.**

Changed.

**P11, L2: 0.95 -> 0.095 um**

Corrected.

**P11, L11: compared to . . .**

Compared to the pure AM-H2SO4 emission scenario (i.e. GEO_AERO_15). Corrected.

**P12, L5: Based on my experience, aerosols are not affecting much on LW fluxes at the surface. This line ("The longwave surface. . .") can be removed.**

Removed.

**P12, L14: "constant climatological SO2"? What does it mean?**

Corrected to: "Kleinschmitt et al. (2018) applied a mean lifetime of 41 days for $SO_2$ to $H_2SO_4$ conversation in their study and found a $SO_2$-to-$H_2SO_4$ conversion rate of 96 %."

**P12, L21: Parenthesis thing - same as in abstract**

Corrected

**P12, L29-30: Just a comment: I don't know has this been pointed out in some earlier studies, but if it has, at least I have missed it. This was an interesting remark and it sounds credible. In addition, the size distribution of particles is different in tropical peak compared to higher latitudes.**

Thank you for the comment. I did not find other studies which made this point in context of solar geoengineering. I added a sentence pointing to the different size distributions in tropical and higher latitudes.

**P13. L18: "...the smaller the negative side effects" Can you really say this? There are several negative side effects which are not studied here.**

I clarified this by adding: "…the smaller the investigated negative side effects…"

**P14, L29: "are only increased by about 4%" I would remove word "only". I was surprised that OH concentration was generally increased.**

Corrected

**Figures: In addition to reviewer 1 comments please correct following typos:**
**Fig3: Areosol -> aerosol (in upper right)**

Corrected

**Fig6: Janauary -> January**

Corrected

---

## Author Comment (AC3) · 22 Feb 2019

**Comments by A. Robock are in bold.** Author responses are in blue.

**I find this sentence in the abstract very confusing: "The modelled all-sky (clear-sky) shortwave radiative forcing for AM-H2SO4 injection scenarios is up to 17-70 % (44-57 %) larger than is the case for SO2." Since you use parentheses the normal way in the rest of the abstract, this usage makes the reader take a long time to figure out what you mean. Please just write this out as, "The modelled all-sky shortwave radiative forcing for AM-H2SO4 injection scenarios is up to 17-70% larger than is the case for SO2, and up to 44-57% larger for the clear-sky case." This takes little more space and is easy to understand. I also have a question about the sentence. How can a value be up to a range of values? Shouldn't it be, "The modelled all-sky shortwave radiative forcing for AM-H2SO4 injection scenarios is up to 70% larger than is the case for SO2, and up to 57% larger for the clear-sky case."? I have not read the rest of the paper yet, but if you use parentheses like this in it, please change the text.**

**See my article:**
**Robock, Alan, 2010: Parentheses are (are not) for references and clarification (saving space). Eos, 91(45), 419, doi:10.1029/2010EO450004.**
**http://climate.envsci.rutgers.edu/robock/Parentheses2010EO450004.pdf**

Dear Dr. Robock,

thank you for your short comment on the abstract. The use of the parentheses was changed in the revised version of the paper and the whole abstract was shortened as well. Furthermore, the expression "up to a range of values" was changed. Instead we refer to average values now.

Yours sincerely,
Sandro Vattioni

---

## Author Comment (AC4) · 22 Feb 2019

**Review of "*Exploring accumulation-mode-H2SO4 versus SO2 stratospheric sulfate geoengineering in a sectional aerosol-chemistry-climate model*" by S. Vattioni et al.**

**Comments by anonymous reviewer #1 are in bold.** Author responses are in blue.

**This article is an interesting comparison between different sulfate geoengineering (SG) injection strategies, namely between $SO_2$ injection (the standard for most simulations up to now) and $H_2SO_4$ (that has been proposed but not as often studied). The study is innovative in its comparison of the two injection strategies (not in the coupling of CCM and sectional aerosol model, as mentioned by the authors at the end of section 2), and I believe that it deserves publication in ACP after the two major issues (and several minor suggestions), that I list below, have been addressed.**

We would like to thank reviewer #1 for her/his insightful comments and suggestions. Our point-by-point replies are given below:

**Major points:**

**1) The authors compare their results with previously obtained results (some of them at least), but they have one major difference with them that is not highlighted anywhere: the longitudinal distribution of the emitted sulfate. Let's take first Niemeier and Timmreck (2015) that the authors mention: they only have one simulation (Geo10-lon) where the injection is spread on all longitudes, and only for a very small latitudinal range (3N to Eq.). In all other scenarios, emissions are only between 120.9 to 123.75 E. Also, Kleinschmitt et al. (2018): they mostly always inject at one single longitude (120E), and only once (BROAD) at 28 locations between 30N and 30S.**

**So, when the authors in the conclusion of their study claim that their RF efficiencies are smaller than those previous ones and they generally point at "lower stratospheric aerosol burdens in our model" they should explain this major difference.**

**I'm not suggesting the authors change their simulations to align closer to previous ones, but the difference between their study and previous ones should be further highlighted, and maybe compared with their sensitivity test (GEO_AERO_point_00 and GEO_SO2_point_00) if they want to make an apple to apple comparison. Overall, I think it would be good that the authors justified much better their assumption that injections over all longitudes are preferable to injections centered in one longitude (and so they chose to do pointed injections only as sensitivity case). Do the authors think this is model dependant? For instance, Tilmes et al (2017) note that, in their model, " Single grid point injections produce sulfate aerosols of smaller size that reflect sunlight more efficiently than injections over a longitude band".**

We agree that many other studies have used point injections at one equatorial grid box. Initially, the focus of our study was to investigate differences of emitting $SO_2$-gas and AM-$H_2SO_4$ as well as the effect of partially emitting outside the tropical pipe. The distribution over all longitudes was mainly chosen to compare our results to Heckendorn et al. (2009), who used the 2D model AER offline and put the results into SOCOL. However, it turned out that the comparison with Heckendorn et al. (2009) is limited as too many parts of the model have changed since their study.

There have been other studies which investigated injections at all longitudes with point emissions only as sensitivity runs (i.e. English et al., 2012, Pierce et al. 2010). As in our study, English et al. (2012) found only small differences between point and longitudinally spread emissions. So far, there are only few studies which investigated pulsed emissions. The opposing behaviour between $SO_2$ and AM-$H_2SO_4$ emission scenarios for sensitivities to temporal and spatial concentrated emissions was first shown in this study.

In our conclusions we changed the scenarios we refer to. We now compare the point emission scenarios to results of Niemeier et al. (2011) and Kleinschmitt et al. (2018). In the rest of the paper we point now to the differences in emission scenarios wherever necessary.

**2) In Table 3 they show what they call "important impacts", naming water vapor and ozone column. However, recent literature has shown that those are not the only (nor the more important) impacts, especially in terms of radiative forcing: what about methane (Visioni et al., 2017 and Tilmes et al., 2018) or ice clouds (Kuebbeler et al., 2012 and Visioni et al, 2018a)? I believe that both factors can't be ignored, when talking about the possible range of impacts of SG. At the very least they should be mentioned, acknowledging previous results, but much better would be to show the changes produced in SOCOL regarding one or both of these aspects. Considering the authors focus also on chemical changes by showing OH and $H_2O$ vertical profiles, I suggest they at least analyse their atmospheric methane changes, comparing them with previous studies.**

There are other side effects of SSG such as tropospheric methane lifetime increase. In our simulation, tropospheric methane mixing ratios remain largely unchanged as we use prescribed mixing ratio boundary conditions for methane at the ground as well as prescribed SST (see figure 1 below). However, the reaction of methane with OH is strongly temperature dependent. In our model we observe a tropospheric temperature decrease of up to 0.95 K which leads to an increase in methane lifetime of up to 2.3 %, while OH concentration almost unchanged in our simulations. Therefore, our model shows a similar effect as in Visioni et al. (2017), but much smaller, as we emitted only 1.83 Mt S per year and Visioni et al. (2017) emitted 5 Mt S per year. When we scale our results linearly to 5 Mt S per year, methane lifetime increases by up to 6.3 % depending on the scenario. This is still less then the 10 % found by Visioni et al. (2017), probably due to the constant SST in our model setup. When taking interactive SST into account, increased changes in temperature, tropospheric ozone and $O^1(D)$ chemistry as well as $H_2O$ concentrations could account for the remaining difference to Visioni et al. (2017). In our simulations, the lifetime of methane at 50 hPa in the lower stratosphere decreases about 14 % in continuous AM-$H_2SO_4$ emission scenarios at all longitudes and about 10 % in the corresponding $SO_2$ emission scenarios, which is in agreement with the OH changes described above.

[Figure]

**Figure 1.** *Vertical profile of the methane volume mixing ratio anomaly. Anomalies represent difference compared to the background simulation. Data show annual and zonal averages between 15° N and 15° S except where indicated. The figure shows results for injections within 10° of the equator as well as zonal averages of January (emissions during January 1ˢᵗ and 2ⁿᵈ) and June (month before emission) from the pulsed simulation.*

Effects on ice clouds cannot be investigated with SOCOL-AER since the model does not include a comprehensive cloud microphysics scheme.

We now clearly state in the text that there are further potential side effects which are not investigated in this study, and discuss the results of the studies mentioned by the referee. We further clarify that Table 3 refers to the negative side effects investigated in this study.

**As I said, before being suitable for publication, I believe a major revision considering these two points is necessary. Furthermore, I have some minor points that I list below:**

**Abstract: The abstract is way too long and confusing. You don't really need to put all your results in the abstract, there's plenty of space elsewhere. Also, the use of the term "surprising" twice is misleading, considering that the last phrase of the abstract is "this study corroborates previous studies". Shorten it by only pointing at some of the results and leave for the discussion all the rest. I agree with Alan Robock that the usage of parenthesis in that way needs to be dropped.**

The abstract has been rewritten and shortened. We now focus on the most important conclusions of the paper, removed the use of the word "surprising" and avoided the overly use of parenthesis. We also adapted the last sentence of the abstract as the part "this study corroborates previous studies" was misleading and weakens new findings made in this paper.

**P. 1, lines 17-19: This phrase is very convoluted and confusing.**

This sentence has been changed.

**P. 2, lines 15-25: As I mentioned in my point 2), are the authors referring to only SG side-effects that reduce the efficiency of the RF produced by the sulfate aerosols (as stated in lines 17-18) or general drawback of SG? I believe the former would be more correct and would make more sense here. Because of this, a bit of clarification is important: (2) of all chemical side effects, ozone depletion is the one with the less significant RF effect (see Pitari et al., 2014). More important would be the effect on methane and other GHGs (see Visioni et al., 2017 and Tilmes et al., 2018) because of changes in photolysis rates and transport. Furthermore, in terms of RF, also the effects of a decrease in UT ice particles, as showed in Kuebbeler et al. (2012) and recently in Visioni et al (2018a), that would produce a cooling effect by trapping less planetary radiation, is important.**

As we write we refer to limitations "*with $SO_2$ injection as a method of producing a radiative forcing perturbation*" like stated in former line 17-18. These limitations include "*effects that reduce the efficiency of the RF produced by the sulfate aerosols*" as stated by reviewer 1 above. We complemented the listing with tropospheric ice clouds and methane and removed ozone depletion from the listing.

**Point (4) has nothing to do with RF and possible effects of SG on ecosystem have been studied in more details after Kravitz et al. (2012) (such as in Xia et al., 2017), together with other changes (tropical storm etc.). But this is not the point of this article.**

Point 4 has been removed.

**P. 4, line 8: Visioni et al (2017, 2018a and 2018b) also used a sectional aerosol approach in their model and fully coupled microphysics and chemistry. It would be good to compare some of their results with yours since they also focus on RF changes, as I explain in point 2) and in some of these comments.**

The three papers you mentioned are now discussed as well and the model which was used is described in one sentence.

**P. 4, line 19: Mills et al. (2017) is more of a description of WACCM-CESM new model set-up and the validation with Pinatubo data, so it is not about SG scenarios.**

Corrected.

**P. 6, lines 11-13: Here the authors should give the reader some more contest on this. The amount injected in this work is not really an issue here (because you are doing sensitivity studies and not looking at the long-term climatic response), but it's a bit of a stretch to say that that's how much Pinatubo injected. We don't know exactly how much SO₂ Pinatubo injected, so from a modeling prespective the amount of SO₂ for Pinatubo is very model dependant (from 10 to 20 Tg-SO₂ to get the best agreement with AOD observations), and it would be good to mention this (Timmreck et al. (2018) would be enough to reference, but just look at, for instance, Mills et al. (2016) where they inject 10 Tg-SO₂ and Pitari et al. (2016), where they inject 20 Tg-SO₂). Just give some contest to the reader. Personally, 1.83 Mt-S per year seems very arbitrary, and I would have preferred an amount more comparable to previous simulations.**

The initial idea was to emit 2 Mt/yr, but due to a problem in the emission scheme, effectively only 1.83 Mt/yr have been emitted. As you already mentioned, this has no effect on the conclusions made in this paper as the same amount of sulfur has been emitted in all our model simulations. The 1.83 Mt/yr correspond to about 9-20% of the amount of S Mt Pinatubo emitted during the eruption 1991 as there are many different estimates ranging from 10-20 Mt S, depending which model one considers.

**P. 7, line 2: I would suggest explaining a bit what the authors mean by "surf zone" and offer some references (as you do, rightfully, for the BDC). It might not be such a common term as you think.**

Was corrected to "…emitting partly into the stratospheric surf zone and not only into the tropical pipe. The stratospheric surf zone is the region outside the subtropical transport barrier where breaking of planetary waves leads to quasi horizontal mixing (McIntyre and Palmer, 1984; Polvani et al., 1995)."

**P. 12, line 1: As per my point 1): most other studies (see Tilmes et al., 2017, but also the entire GeoMIP G4 experiment) inject at only one longitude. In other simulations the single longitude was found either better for the scope ($r_{eff}$ closer to the desired one, Tilmes et al., 2017) or unimportant (because of the fast mixing time). You should mention this here.**

We refer now to other studies which have previously shown that point injections result in larger aerosol burden, $r_{eff}$ closer to the desired size and larger short-wave radiative forcing values, i.e. Niemeier et al. (2011) and Tilmes et al., (2017).

**P. 12, line 5: The shared longwave surface anomaly doesn't really say anything, especially since you don't have a surface coupling. It would not be due to the aerosol absorption anyway, but it is the result of many more processes. I suggest removing this phrase or explain better what you think the surface LW anomaly tells you, if you think it's important.**

The sentence about the long-wave anomaly has been removed.

**P. 12, line 14: constant climatological what? Concentration? It's confusing.**

Corrected to: "Kleinschmitt et al. (2018) applied a mean lifetime of 41 days for $SO_2$ to $H_2SO_4$ conversation in their study and found a $SO_2$-to-$H_2SO_4$ conversion rate of 96 %."

**P. 12, line 20: split this in two phrases, one for AS and one for CS. Don't use parenthesis this way.**

Corrected. I split the sentence into two.

**P. 13, line 5: Either you write "Emission strategies like that investigated by …" or "Emitting at … as investigated by …"**

The sentence has been rewritten.

**P. 13, line 16-23: Again: ozone and water vapor are only two of the SG side effects when it comes to RF: you should at least mention the fact that methane lifetime would increase and ice cloud decrease.**

We now explicitly state that only the negative side effects investigated in this study are considered in Table 3. Additionally, we looked at changes in methane lifetime, and point to other side effects (i.e. cloud feedbacks) which could have an inhibiting effect on solar geoengineering efficiency but are not considered here.

**P. 14, line 1: like in the case of $SO_2$ emissions**

Corrected.

**P. 15, lines 7-8: as I said in my major point 1), mention the big difference between the studies, that is the different longitudinal distribution of the injections.**

We now compare our point emission scenarios to Niemeier et al. (2011) and Kleinschmitt et al. (2018) and point out the difference between the studies.

**P. 15, lines 13-14: Yes, but in the framework of CCMI, so considering reference scenarios. The response to the BDC to the stratospheric heating produced by SG is not something that is considered there. The reference is still useful but mention this, at least.**

Corrected. I have added a sentence for clarification.

**P. 15, line 24: As has been previously studied in Kuebbeler et al (2012) and Visioni et al (2018a).**

References have been added.

**P. 16, line 6: "with" current GCMs.**

Corrected

**Figure 2: This figure has a very poor resolution (compared to Fig. 5 that is similar). Panels b) and c) have such large scales and the curves are minuscule.**

The resolution of all figures has been increased and the scale of panels b) and c) of figure 2 and panel b) of figure 5 has been adapted.

**Figure 4: This is an interesting figure but must be improved. The scale for the SW forcing could be reduced a lot to higlight the differences between the different simulations (panel b from -1 to .3 W/m2 and panel c from 0 to -2 W/m2). Furthermore, symbols are very hard to read. I suggest enlarging them (or using more color). Very minor thing, the is and ls in figure 4 are all "weird", they look bold while the rest of the letters don't. You should fix this. (same goes for figure 2, 6 and 7). It's probably just a problem of how you saved the figures.**

The scale was adapted, and symbols were enlarged. The weird looking letters were replaced as well.

**References:**

English, J. M., Toon, O. B. and Mills, M. J.: Microphysical simulations of sulfur burdens from stratospheric sulfur geoengineering, Atmospheric Chem. Phys., 12(10), 4775–4793, doi:10.5194/acp-12-4775-2012, 2012.

Heckendorn, P., Weisenstein, D., Fueglistaler, S., Luo, B. P., Rozanov, E., Schraner, M., Thomason, L. W. and Peter, T.: The impact of geoengineering aerosols on stratospheric temperature and ozone, Environ. Res. Lett., 4(4), doi:Artn 045108 10.1088/1748-9326/4/4/045108, 2009.

Pierce, J. R., Weisenstein, D. K., Heckendorn, P., Peter, T. and Keith, D. W.: Efficient formation of stratospheric aerosol for climate engineering by emission of condensible vapor from aircraft, Geophys. Res. Lett., 37(18), doi:10.1029/2010GL043975, 2010.

Kleinschmitt, C., Boucher, O., and Platt, U.: Sensitivity of the radiative forcing by stratospheric sulfur geoengineering to the amount and strategy of the $SO_2$ injection studied with the LMDZ-S3A model, Atmos. Chem. Phys., 18, 2769-2786, https://doi.org/10.5194/acp-18-2769-2018, 2018.

Kuebbeler, M., Lohmann, U., and Feichter, J.: Effects of stratospheric sulfate aerosol geo-engineering on cirrus clouds, Geophys. Res. Lett., 39, L23803, https://doi.org/10.1029/2012GL053797, 2012.

Niemeier, U. and Timmreck, C.: What is the limit of climate engineering by stratospheric injection of $SO_2$?, Atmos. Chem. Phys., 15, 9129-9141, https://doi.org/10.5194/acp-15-9129-2015, 2015.

Pitari, G., Di Genova, G., Mancini, E., Visioni, D., Gandolfi, I., and Cionni, I.: Stratospheric Aerosols from Major Vol- canic Eruptions: A Composition-Climate Model Study of the Aerosol Cloud Dispersal and e-folding Time, Atmosphere, 7, 79, https://doi.org/10.3390/atmos7060075, 2016.

Tilmes, S., J. H. Richter, M. J. Mills, B. Kravitz, D.G. MacMartin, F. Vitt, J. J. Tribbia, and J.-F. Lamarque, 2017: Sensitivity of aerosol distribution and climate response to stratospheric SO2 injection locations, JGR-Atmospheres

Tilmes, S., Richter, J. H., Mills, M. J., Kravitz, B., MacMartin, D. G., Garcia, R. R., et al. 2018: Effects of different stratospheric $SO_2$ injection altitudes on stratospheric chemistry and dynamics. *Journal of Geophysical Research: Atmospheres*, 123, 4654–4673

Timmreck, C., Mann, G. W., Aquila, V., Hommel, R., Lee, L. A., Schmidt, A., Brühl, C., Carn, S., Chin, M., Dhomse, S. S., Diehl, T., English, J. M., Mills, M. J., Neely, R., Sheng, J., Toohey, M., and Weisenstein, D.: The Interactive Stratospheric Aerosol Model Intercomparison Project (ISA-MIP): motivation and experimental design, Geosci. Model Dev., 11, 2581-2608, https://doi.org/10.5194/gmd-11-2581-2018, 2018.

Visioni, D., Pitari, G., Aquila, V., Tilmes, S., Cionni, I., Di Genova, G., and Mancini, E.: Sulfate geoengineering impact on methane transport and lifetime: results from the Geoengineering Model Intercomparison Project (GeoMIP), Atmos. Chem. Phys., 17, 11209-11226, https://doi.org/10.5194/acp-17-11209-2017, 2017.

Visioni, D., Pitari, G., di Genova, G., Tilmes, S., and Cionni, I.: Upper tropospheric ice sensitivity to sulfate geoengineering, Atmos. Chem. Phys., 18, 14867-14887, https://doi.org/10.5194/acp-18-14867-2018, 2018a.

Visioni, D., Pitari, G., Tuccella, P., and Curci, G.: Sulfur deposition changes under sulfate geoengineering conditions: quasi-biennial oscillation effects on the transport and lifetime of stratospheric aerosols, Atmos. Chem. Phys., 18, 2787-2808, https://doi.org/10.5194/acp-18-2787-2018, 2018b.

Xia, L., Nowack, P. J., Tilmes, S., and Robock, A.: Impacts of stratospheric sulfate geoengineering on tropospheric ozone, Atmos. Chem. Phys., 17, 11913-11928, https://doi.org/10.5194/acp-17-11913-2017, 2017.